# Global disparities in the introduction, scale-up, and effectiveness evaluation of COVID-19 vaccines

Martina Pesce [1], Daniel R. Feikin [2], Melissa M. Higdon [3],
Katherine L. O'Brien[2], Minal K. Patel[2,4], Analía Rearte [5,6], Carla Vizzotti[6],
Annelies Wilder-Smith[2] & Edward P. K. Parker [1,7]

The global response to COVID-19 saw the most rapid and extensive vaccination rollout in history. Yet there were large disparities in the introduction, scale-up, and evaluation of programmes. To systematically quantify these disparities, we generate linkages across public datasets containing country- and territory-level income data, COVID-19 vaccination rates, and COVID-19 vaccine effectiveness (VE). Our results show that, compared with high-income countries, lower-income countries introduced vaccines later, were less likely to achieve key coverage milestones, and were slower to do so where these milestones were achieved. The literature on primary series COVID-19 VE has been dominated by studies of mRNA vaccines from high-income countries, with data for other vaccines and lower-income countries appearing later and in substantially lower quantities. For vaccines with available VE data across multiple income settings (BNT162b2, mRNA-1273, and ChAdOx1-S), our meta-regression highlights robust protection against severe COVID-19, with no significant differences in primary series VE according to country-level income status during the Delta and Omicron periods. Our findings demonstrate the strong protection conferred by COVID-19 vaccines across diverse populations. Nonetheless, our results quantify the stark disparities that pervaded each stage of COVID-19 vaccine implementation, and highlight evidence gaps related to products and platforms being used across much of the globe.

Following the onset of the COVID-19 pandemic, vaccines capable of reducing the risk of SARS-CoV-2 infection, disease, and severe outcomes were developed at unprecedented speed and introduced worldwide from December 2020[1,2]. The ensuing vaccine rollout was characterised by marked global disparities in scale and timing, with low- and middle-

income countries (LMICs) facing significant delays in vaccine access compared to high-income countries (HICs)[3,4].

Following implementation, the robust assessment of vaccine effectiveness (VE) is crucial to provide evidence that vaccines are effective in real-world settings, including against severe COVID-19 and

[1]The Vaccine Centre, London School of Hygiene & Tropical Medicine, London, UK. [2]Department of Immunization, Vaccines and Biologicals, World Health Organization, Geneva, Switzerland. [3]International Vaccine Access Center, Department of International Health, Johns Hopkins Bloomberg School of Public Health, Baltimore, MD, USA. [4]U.S. Public Health Service Commissioned Corps, Rockville, MD, USA. [5]Escuela de Medicina, Universidad Nacional de Mar del Plata, Mar del Plata, Argentina. [6]Universidad Nacional de San Martín, Buenos Aires, Argentina. [7]Department for Infectious Disease Epidemiology and International Health, London School of Hygiene & Tropical Medicine, London, UK. ✉e-mail: edward.parker@lshtm.ac.uk

in response to emerging variants. Obtaining timely VE estimates across diverse populations is important to support local and international policy decisions and to promote confidence in vaccination among populations underrepresented in pre-licensure trials. The ability of a country to conduct VE studies depends on having access to vaccines, surveillance systems, and sufficient human and financial resources[3]. VE estimates are affected by the speed, coverage, and schedule of vaccine implementation. Protection may also vary according to demographic and clinical characteristics of the vaccinated population, and reduced immune responses and efficacy have been reported in LMICs relative to HICs for multiple vaccines[5–8]. This may limit the generalisability of VE findings across geographically disparate settings.

In this study, we harnessed linkages across large public datasets to systematically quantify global disparities in the implementation and real-world VE evaluation of COVID-19 vaccines. Using a harmonised analysis approach, we considered the association between country-level income status and (i) vaccine introduction; (ii) vaccine scale-up; (iii) the publication of VE studies and (iv) VE against severe disease.

## Results
### Global differences in vaccine introduction and scale-up
To explore milestones in vaccine introduction and scale-up, we linked World Bank country- and territory-level income data to vaccine coverage databases maintained by the World Health Organization (WHO)[9] and Our World in Data (OWID)[4]. These databases collate estimates of coverage per capita from health ministries, government websites, direct reports to WHO, and third-party sources. We defined introduction as the earliest recorded vaccination date in WHO or OWID. Among 204 countries and territories with World Bank data on per capita Gross National Income (GNI), 203 had records of COVID-19 vaccine introduction as of 31 December 2021 (Figs. 1A, 2A and Supplementary Table 1; see Supplementary Data 1 for full dataset). The median vaccine introduction date—reflecting the point at which 50% of countries and territories had recorded at least one vaccination—was 30 December 2020 in GNI quartile 4 (highest income), 26 January 2021 in quartile 2, 24 February 2021 in quartile 3, and 15 March 2021 in quartile 4 (lowest income; Fig. 2B; see Supplementary Table 1 for interquartile ranges [IQRs]). A strong correlation between income status and vaccine introduction date was also apparent when handling GNI as a continuous variable (Spearman's rho −0.66, p < 0.0001; 203 estimates; Fig. 3A). When comparing WHO and OWID databases, 187/203 (92%) countries and territories had records of vaccine introduction in both databases, of which 108/187 (58%) dates were within ±7 days (median [IQR] difference of 6 [2–16] days).

We then explored the scale-up of vaccine implementation based on several coverage milestones specified in the 2021 WHO global COVID-19 vaccination strategy[10]. The milestone of 40% primary series coverage by 31 December 2021 (with 'primary series' defined by the product-specific use authorisation) was reached by 47/51 (92%), 39/51 (77%), 20/51 (39%), and 3/50 (6%) countries and territories in GNI quartiles 4 (highest income), 3, 2, and 1 (lowest income), respectively (Figs. 1B and 2A; Supplementary Table 1). Among countries and territories that achieved 40% coverage (either before or after the milestone target of 31 December 2021), the median (IQR) time from vaccine introduction to this coverage threshold was 190 (160–209), 239 (182–274), 272 (224–356), and 450 (347–637) days, respectively (Fig. 2B). Clear discrepancies among GNI quartiles were also apparent when restricted to countries and territories achieving 40% coverage before 31 December 2021 (Supplementary Table 1).

The subsequent milestone of 70% coverage by 30 June 2022 was reached by 36/51 (71%), 19/51 (37%), 7/51 (14%), and 5/50 (10%) countries and territories in GNI quartiles 4 (highest income), 3, 2, and 1 (lowest income), respectively. The median (IQR) time from vaccine introduction to 70% coverage was 268 days (240–348), 384 (306–418), 379 (326–501), and 579 (457–619) days, respectively. Disparities in vaccine

scale-up were also apparent when considering GNI as a continuous variable (Spearman's rho −0.73, p < 0.0001; 146 estimates; Fig. 3B) and when restricted to countries and territories achieving 70% coverage by 30 June 2022 (Supplementary Table 1).

In the context of limited supply, vaccinating high-risk groups such as older adults and healthcare workers was recommended as part of the WHO roadmap for COVID-19 vaccine prioritisation[11]. Based on data collated by the WHO/UNICEF[12], median (IQR) coverage in healthcare workers as of December 2021 was 76% (66–96%), 79% (50–95%), 97% (58–100%), and 59% (31–100%) in GNI quartiles 4 (highest income), 3, 2, and 1 (lowest income), respectively (Fig. 4). Notably, data on coverage in healthcare works were available for 122/203 (60%) countries and territories (see Supplementary Table 2 for breakdown by GNI quartile), and among those reporting, the median (IQR) proportion of the population accounted for by healthcare workers varied from 5.4% (3.4–6.9%) in GNI quartile 4 (highest income) to 0.4% (0.3–0.8%) in GNI quartile 1 (lowest income).

Data on coverage in older adults as of December 2021 were available for 127/203 (63%) countries and territories. Where defined, the threshold used to define older adults varied between 50 and 75 years, with a skew towards younger thresholds in lower-income quartiles (Supplementary Table 3). Among countries and territories reporting, the median (IQR) proportion of the total population accounted for by older adults varied from 25% (17–27%) in GNI quartile 4 (highest income) to 9% (8–11%) in GNI quartile 1 (lowest income; Supplementary Table 2). Median (IQR) coverage in older adults was 89% (75–96%), 71% (63–85%), 50% (32–72%), and 10% (4–27%) in GNI quartiles 4 (highest income), 3, 2, and 1 (lowest income), respectively. Disparities in vaccine coverage among healthcare workers and older adults across GNI quartiles were smaller than the disparities observed in population-wide coverage (Fig. 4 and Supplementary Table 2). As of December 2023 (marking the end of COVAX, the multi-partner collaboration to promote the development, manufacture, fair allocation, and delivery of COVID-19 vaccines), coverage rates in high-risk groups had increased across income strata, with the most marked changes in GNI quartiles 1 and 2 (Fig. 2 and Supplementary Table 2).

Among 25 vaccine products in use globally, vectored vaccines were the most widely deployed (187/203 [92%] countries and territories), followed by mRNA vaccines (152/203 [75%]), inactivated vaccines (119/203 [59%]), and protein vaccines (41/203 [20%]), although the relative share of doses for each platform is not reliably captured based on available data. Distributions by GNI quartile are provided by platform (Fig. 2C) and product (Supplementary Fig. 1); proportionately, mRNA vaccines were most utilised in GNI quartile 4 (highest income) and least utilised in GNI quartile 1 (lowest income).

### Global differences in the evaluation of vaccine effectiveness
We next sought to systematically quantify disparities in the availability of VE data for different products and platforms. We obtained studies of absolute primary series VE—which measures the reduction in SARS-CoV-2 infection or disease among vaccinated compared to unvaccinated individuals—from published, preprint, and grey literature sources using the VIEW-hub living literature review conducted by the Johns Hopkins School of Public Health International Vaccine Access Center (IVAC)[13]. The VIEW-hub database collates VE studies meeting pre-specified criteria of methodological rigour, including adequate accounting for confounding variables and inclusion of laboratory-confirmed outcomes.

As of 11 January 2024, the database contained 425 studies with country- or territory-specific VE estimates, of which 423 had corresponding GNI and coverage data (Fig. 1C and Supplementary Fig. 2A; see Supplementary Data 2 for full dataset). Of these, 345 (82%), 64 (15%), 9 (2%) and 5 (1%) were from GNI quartiles 4 (highest income), 3, 2, and 1 (lowest income), respectively. Among 203 countries and territories reporting vaccine introduction, VE data were available from 48 (24%), representing 24/51 (47%), 15/51 (29%), 7/51 (14%), and 2/50 (4%)

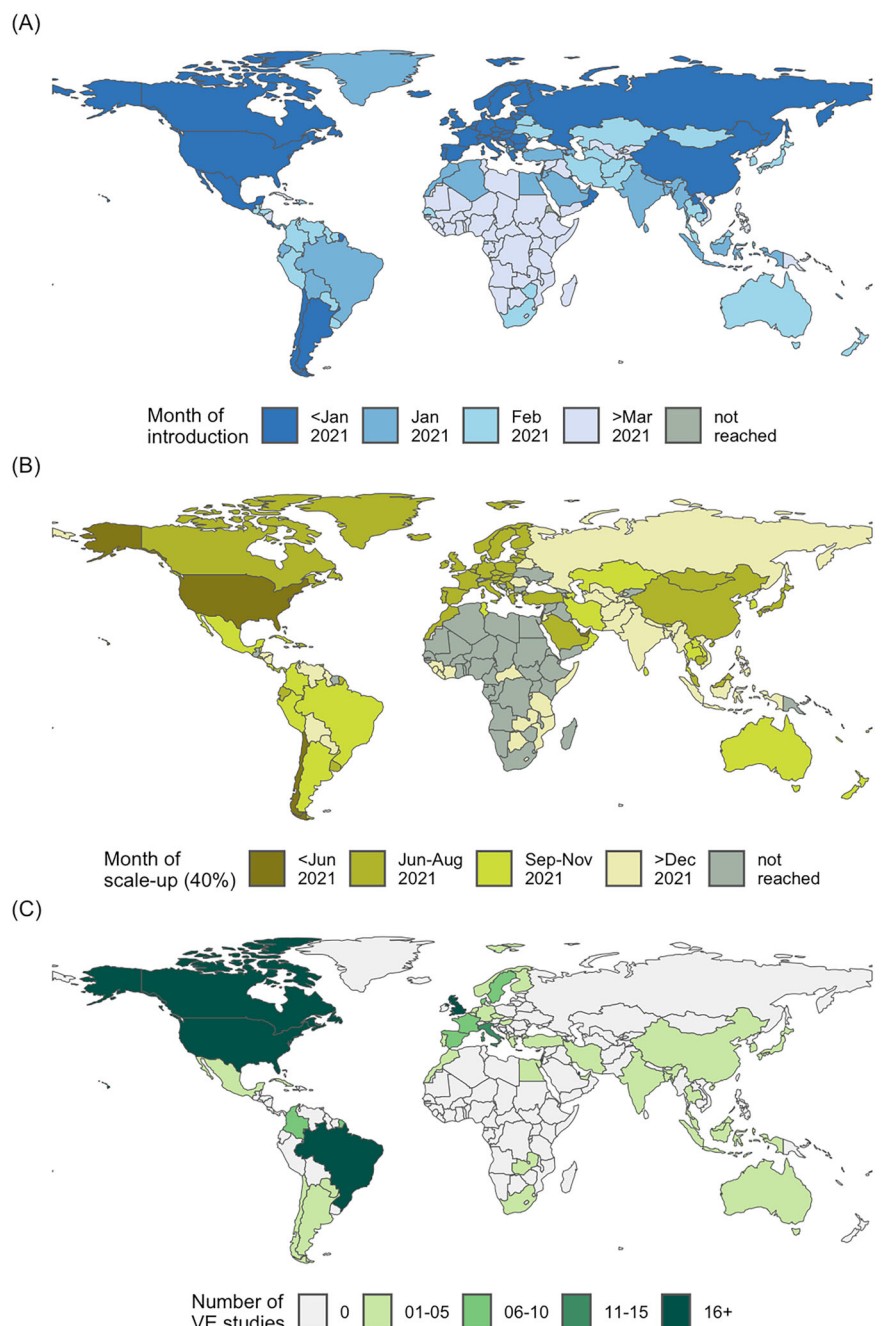

**Fig. 1 | Global disparities in COVID-19 vaccine introduction, scale-up, and effectiveness evaluation.** Maps of **A** COVID-19 vaccine introduction dates, **B** scale-up to 40% population-wide primary series coverage, and **C** distribution of available VE studies across different countries and territories. VE vaccine effectiveness.

in GNI quartiles 4, 3, 2, and 1, respectively (Fig. 2A). Collectively, the USA (135 [32%]), the UK (48 [11%]), Israel (30 [7%]), Canada (28 [7%]), and Brazil (17 [4%]) accounted for 258 (61%) of the 423 VE studies included.

The earliest publication date for VE data in GNI quartile 4 (highest income) was 18 February 2021[14,15]. VE data for GNI quartiles 3, 2, and 1 were first published on 07 July 2021[16], 27 January 2022[17], and 23 November 2021[18], respectively, although studies from quartile 4 continued to predominate throughout the course of the pandemic (Supplementary Fig. 3). Among countries and territories with at least one published VE estimate, the median (IQR) time from vaccine introduction to the first VE research publication was 219 days (125–466), 354 (316–460), 411 (390–678), and 366 (338–393) for GNI quartiles 4 (highest income), 3, and 2, respectively. Consistent with these

disparities, there was a strong correlation between GNI as a continuous variable and the date of first VE research publication (Spearman's rho −0.42, p = 0.003; 48 estimates; Fig. 3C).

Considering data by vaccine platform, 376/423 (89%) studies reported on mRNA vaccines (earliest on 18 February 2021[14,15]), 128 (30%) reported on vectored vaccines (earliest on 02 March 2021[19]), 65 (15%) reported on inactivated vaccines (earliest on 07 July 2021[16]), and 1 (0.2%) reported on protein vaccines (on 05 April 2022[20]; Fig. 2D). Most primary series VE studies reported on the Delta (160 [38%]) or Omicron (152 [36%]) variants (Table 1). Studies reported VE estimates for outcomes of varying severity (with individual studies often reporting separate estimates for multiple outcomes). The most frequent outcome was infection (including test positivity irrespective of symptom status; 255 [60%]), followed by severe COVID-19 (241 [57%]),

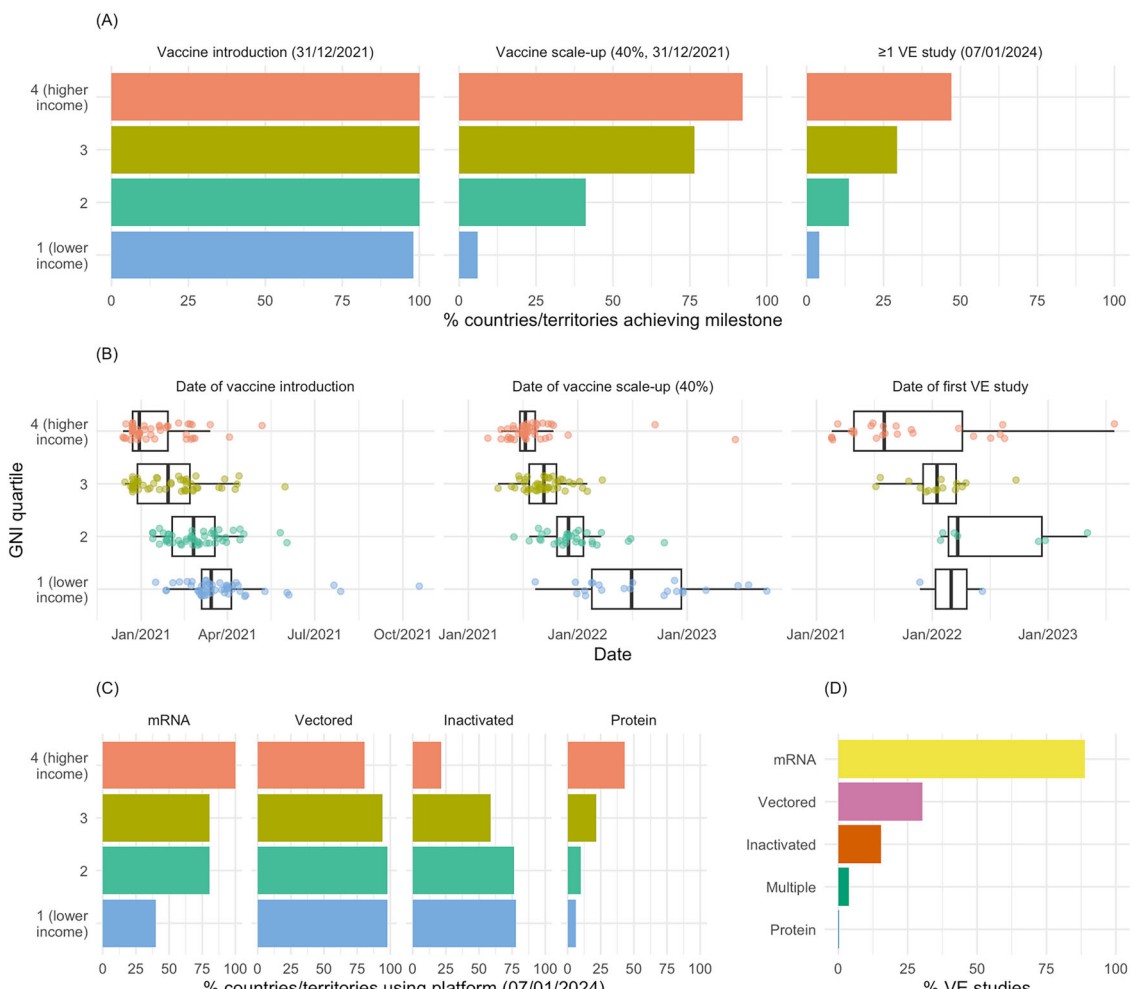

**Fig. 2 | Disparities in COVID-19 vaccine introduction, scale-up, and effectiveness evaluation by income quartile. A** Percentage of countries and territories achieving vaccine introduction by 31 December 2021 (left), scale-up to 40% population-wide primary series coverage by 31 December 2021 (middle), and publication of ≥1 VE study by 07 January 2024 (right) by GNI quartile (n countries/territories = 51, 51, 51, and 50 for quartiles 4, 3, 2 and 1, respectively). **B** Date of vaccine introduction (left; n = 51, 51, 51, and 50 for quartiles 4, 3, 2, and 1, respectively), scale-up to 40% population-wide primary series coverage (middle; n = 49, 43, 32, and 22) and first VE study publication (right; n = 24, 15, 7, and 2) by GNI

quartile. Only countries and territories that reached the specified milestones are included. **C** Percentage of countries and territories using different vaccine platforms by GNI quartile as of 07 January 2024 (n = 51, 51, 51, and 50 for quartiles 4, 3, 2, and 1, respectively). **D** Percentage of primary series VE studies by platform as of 11 January 2024 (n studies = 423). Box plots display median (centre line), upper and lower quartiles (box limits), the minimum value greater than or equal to the lower quartile − 1.5 × interquartile range (lower whisker), and the largest value less than or equal to the upper quartile + 1.5 × interquartile range (upper whisker). GNI per capita Gross National Income, VE vaccine effectiveness.

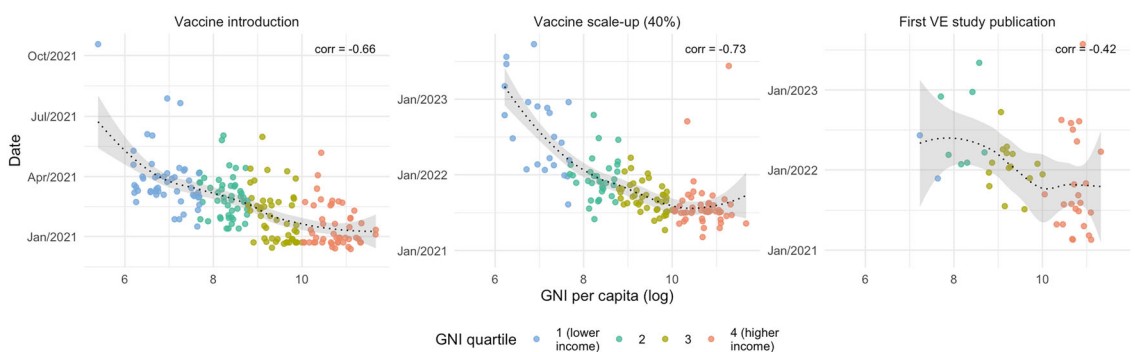

**Fig. 3 | Correlation between COVID-19 vaccine implementation metrics and per capita Gross National Income.** Per capita GNI was defined based on World Bank estimates using the Atlas method. Vaccine introduction and scale-up data were obtained from public repositories maintained by the WHO and Our World in Data. VE data were obtained on 11 January 2024 from VIEW-hub. Spearman's rank correlation statistics were −0.66 for vaccine introduction (p = 1.31 × 10⁻²⁶; 203

estimates), −0.73 for vaccine scale-up (p = 1.61 × 10⁻²⁵; 146 estimates), and −0.42 for first VE research publication (p = 0.0027; 48 estimates). Lines show local weighted regression (LOESS) fits with 95% confidence intervals. Spearman's rank correlation coefficients were calculated using two-sided tests. corr, Spearman's rank correlation coefficient, VE vaccine effectiveness.

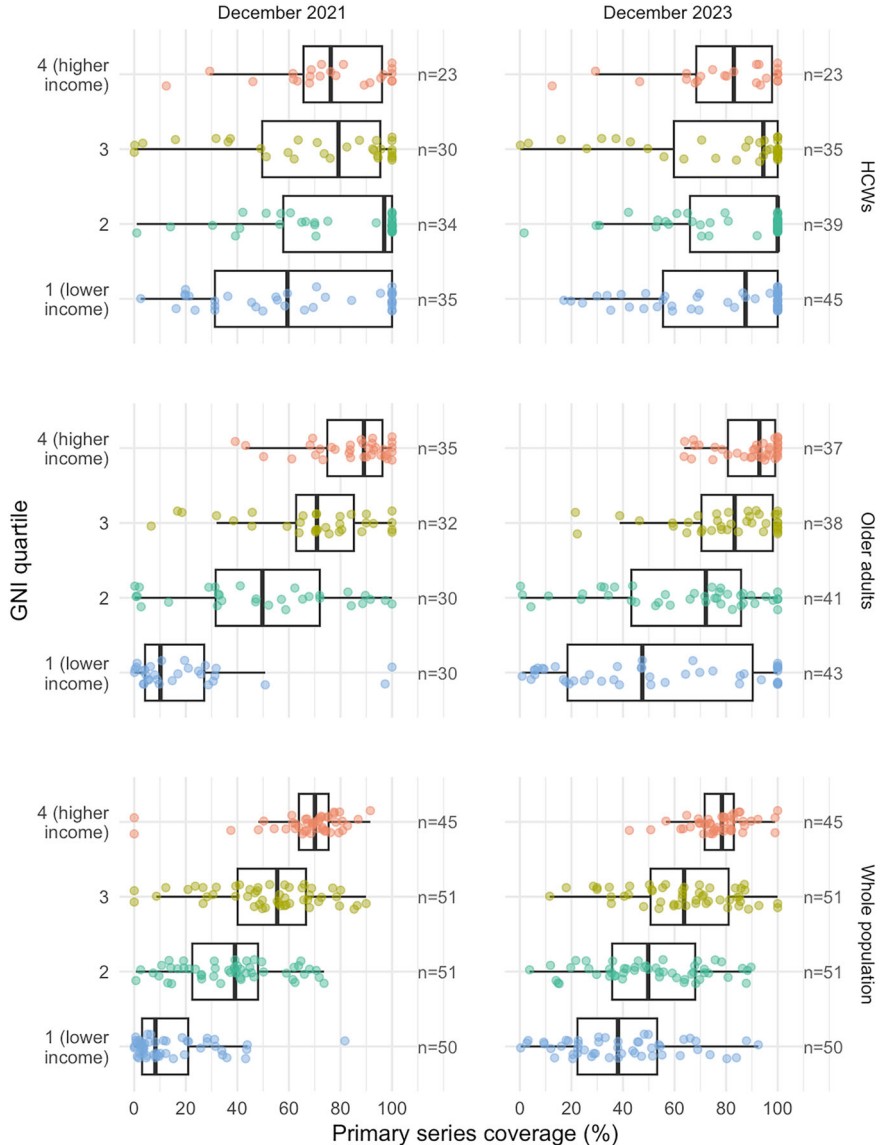

**Fig. 4 | Vaccine coverage in high-risk populations (healthcare workers and older adults) and overall.** Primary series coverage estimates as of December 2021 and December 2023 were obtained from WHO/UNICEF. Vaccination data collection methods and denominator inclusion criteria for healthcare workers were not defined in publicly available data and may have varied across countries and territories. The age threshold to define older adults varied by country and territory (Supplementary Table 3). Whole population denominators reflect estimates of overall population size by the United Nations Population Division. Box plots display median (centre line), upper and lower quartiles (box limits), the minimum value greater than or equal to the lower quartile − 1.5 × interquartile range (lower whisker), and the largest value less than or equal to the upper quartile + 1.5 × interquartile range (upper whisker). HCWs healthcare workers, GNI per capita Gross National Income.

symptomatic disease (142 [34%]), and COVID-19-related mortality (73 [17%]). The earliest publication date for VE against severe COVID-19 was 18 February 2021 for mRNA vaccines[15], 17 March 2021 for vectored vaccines[21], 07 July 2021 for inactivated vaccines[16], and 05 April 2022 for protein vaccine[20].

## Vaccine effectiveness by income status

From a global policy perspective, disparities in VE data availability are particularly problematic if findings from high-income settings are not generalisable to low-income settings—e.g. due to differences in demographic and clinical factors that impact the strength and duration of vaccine responses[5–8], comorbidity rates, vaccine eligibility criteria, social mixing patterns, and non-pharmaceutical interventions. The availability of data from different GNI quartiles provides an opportunity to explore potential geographic differences in VE based on the emerging global evidence base. To this end, we performed a

meta-regression to assess variation in VE by country- and territory-level income status. We selected product- and variant-specific VE estimates relating to adults (of any age group), complete homologous primary series vaccination, and with representation across multiple GNI quartiles. Outcomes reported in the literature included infection, symptomatic COVID-19, severe COVID-19, and COVID-19-related deaths. After excluding estimates of infection and mild symptomatic disease due to significant variation in testing recommendations and practices across countries, our outcomes of interest were severe COVID-19 and COVID-19-related mortality, although no comparisons of the latter were eligible for inclusion (Supplementary Table 4). A total of 51 studies met the eligibility criteria for inclusion in our meta-regression (Supplementary Data 2). These included studies of BNT162b2 (n = 42), mRNA-1273 (n = 26), and ChAdOx1-S (n = 22) reporting on VE against severe COVID-19 due to the Delta and Omicron variants (Supplementary Table 5). Other products and variants lacked

**Table 1 | Characteristics of primary series vaccine effectiveness studies reporting on country- or territory-specific estimates**

| | | Total (N = 423) | mRNA (N = 376) | Vectored (N = 128) | Inactivated (N = 65) | Multiple (N = 16) | Protein (N = 1) |
|---|---|---|---|---|---|---|---|
| Variant platform[a] | mRNA | 376 (89%) | – | – | – | – | – |
| | Vectored | 128 (30%) | – | – | – | – | – |
| | Inactivated | 65 (15%) | – | – | – | – | – |
| | Multiple | 16 (4%) | – | – | – | – | – |
| | Protein | 1 (0%) | – | – | – | – | – |
| Vaccines[a] | BNT162b2 | 256 (61%) | 256 (68%) | – | – | – | – |
| | mRNA-1273 | 104 (25%) | 104 (28%) | – | – | – | – |
| | Multiple or hetero-logous RNA | 139 (33%) | 139 (37%) | – | – | – | – |
| | ChAdOx1-S | 100 (24%) | – | 100 (78%) | – | – | – |
| | Ad26.COV2.S | 45 (11%) | – | 45 (35%) | – | – | – |
| | Gam-COVID-Vac/Sputnik V | 11 (3%) | – | 11 (9%) | – | – | – |
| | Ad5-nCoV | 5 (1%) | – | 5 (4%) | – | – | – |
| | CoronaVac | 48 (11%) | – | – | 48 (74%) | – | – |
| | BBIBP-CorV | 19 (4%) | – | – | 19 (29%) | – | – |
| | BBV152 | 6 (1%) | – | – | 6 (9%) | – | – |
| | COVIran | 1 (0%) | – | – | 1 (2%) | – | – |
| | Abdala | 1 (0%) | – | – | – | – | 1 (100%) |
| | Heterologous platforms/comparative | 14 (4%) | 1 (0%) | – | – | 16 (100%) | – |
| Variant[a] | Delta | 160 (38%) | 142 (38%) | 51 (40%) | 19 (29%) | 8 (50%) | 1 (100%) |
| | Omicron | 152 (36%) | 143 (38%) | 24 (19%) | 23 (35%) | 3 (19%) | – |
| | Alpha | 76 (18%) | 69 (18%) | 29 (23%) | 4 (6%) | 1 (6%) | – |
| | Original | 13 (3%) | 12 (3%) | 2 (2%) | 1 (2%) | – | – |
| | Gamma | 12 (3%) | 6 (2%) | 7 (5%) | 3 (5%) | 1 (6%) | – |
| | Beta | 10 (2%) | 9 (2%) | 2 (2%) | – | – | – |
| | Mu | 4 (1%) | 4 (1%) | 3 (2%) | 4 (6%) | – | – |
| | Multiple[b] | 158 (37%) | 141 (38%) | 57 (45%) | 20 (31%) | 7 (%) | – |
| | Unknown | 3 (1%) | 3 (1%) | – | – | – | – |
| Outcome[a] | Infection[c] | 255 (60%) | 232 (62%) | 66 (52%) | 29 (45%) | 10 (62%) | – |
| | Symptomatic | 142 (34%) | 126 (34%) | 39 (30%) | 17 (26%) | 5 (31%) | – |
| | Severe | 241 (57%) | 205 (55%) | 80 (62%) | 49 (75%) | 8 (50%) | 1 (100%) |
| | Death | 73 (17%) | 56 (15%) | 33 (26%) | 28 (43%) | 1 (6%) | 1 (100%) |
| GNI quartile | 1 (lower income) | 5 (1%) | – | 4 (3%) | 2 (3%) | – | – |
| | 2 | 9 (2%) | 5 (1%) | 5 (4%) | 8 (12%) | – | – |
| | 3 | 64 (15%) | 36 (10%) | 32 (25%) | 41 (63%) | 4 (25%) | 1 (100%) |
| | 4 (higher income) | 345 (82%) | 335 (89%) | 87 (68%) | 14 (22%) | 12 (75%) | – |

Data are n studies (%). GNI per capita Gross National Income.
[a]individual studies could report on multiple groups; percentages may therefore sum to more than 100%.
[b]studies assessed vaccine effectiveness over a period spanning multiple variants.
[c]includes asymptomatic disease and SARS-CoV-2 test positivity (irrespective of symptom status).

sufficient representation across GNI quartiles (Supplementary Fig. 2B). GNI quartile 4 (highest income) accounted for 38/51 (75%) of the included studies, GNI quartile 3 for 9/51 (18%), while GNI quartiles 2 and 1 (lowest income) each contributed 2/51 (4%) studies.

For BNT162b2, pooled VE estimates for primary (2-dose) vaccination against severe COVID-19 due to the Delta variant were 94% (95% CI 89–96%, 19 studies, 9 countries/territories), 91% (95% CI 81–95%, 6 studies, 6 countries/territories) and 92% (95% CI 29–99%, 1 study, 1 country/territory) for GNI quartiles 4, 3, and 2, respectively (Fig. 5). The median (IQR) for the upper limit of follow-up (the maximum potential duration of follow-up among study participants) after primary series vaccination was 25 (21–35) and 28 (16–48) weeks for GNI quartiles 4 and 3, respectively, and 38 weeks for the single estimate from GNI quartile 2. We observed no significant difference in VE across GNI quartiles (p = 0.7074). Of the observed variation in VE estimates not attributable to sampling error (quantified via the I² statistic), 55% was attributed to

differences between countries/territories (I² level 3), while heterogeneity within countries/territories accounted for 43% (I² level 2). Pooled VE estimates for severe COVID-19 due to the Omicron variant were 74% (95% CI 57–84%, 18 studies, 6 countries/territories), 67% (95% CI 38–82%, 5 studies, 4 countries/territories), and 50% (95% CI −49–87%, 1 study, 1 country/territory) for GNI quartiles 4, 3, and 2, respectively (p = 0.6284 for test of differences among quartiles). Again, there was no systematic variation in the upper limit of follow-up across quartiles, and 79% of variation across VE estimates was attributed to heterogeneity between countries/territories (Supplementary Table 6).

For each vaccine evaluated (BNT162b2, mRNA-1273, and ChAdOx1-S), we observed no significant VE differences according to GNI quartile (Fig. 5 and Supplementary Table 6). Pooled VE estimates were similar across GNI quartiles, with deviations attributable to single-country estimates with wide confidence intervals. Our findings were consistent in a sensitivity analysis limited to VE estimates with a

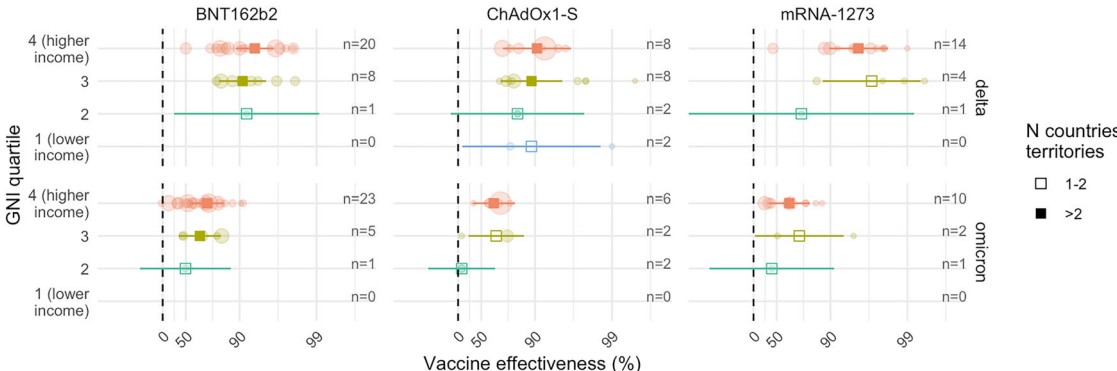

**Fig. 5 | Meta-regression of primary series vaccine effectiveness against severe COVID-19 by variant context and income status.** The squares and error bars indicate meta-analytical pooled estimates of vaccine effectiveness for each GNI quartile with 95% confidence intervals. Open squares indicate estimates in which there were 1–2 countries/territories represented for a given quartile. Each point represents an individual vaccine effectiveness estimate. Point size is proportional to study weight on an inverse variance scale, such that smaller points reflect smaller weights in the meta-regression. GNI quartile was included as a moderator in meta-analytic models, while country/territory was included as a random effect to account for the anticipated clustering of estimates at country/territory level. The absence of data for GNI quartile 1 for all but one comparison reflects the absence of estimates for this quartile that met the meta-regression inclusion criteria. The number of VE estimates is displayed to the right-hand side of each plot. GNI per capita Gross National Income.

maximum upper limit of follow-up of 26 weeks (Supplementary Table 7). See Supplementary Materials for citation details and Supplementary Fig. 4–6 for forest plots.

## Discussion

We used data linkages to systematically quantify global disparities in the implementation, scale-up, and evaluation of COVID-19 vaccines. The median time to vaccine introduction was over 70 days longer for countries in the lowest versus the highest GNI quartile. After introduction, countries in the lowest GNI quartile were less likely to achieve scale-up to 40% primary series coverage and—where this threshold was achieved—were a median of more than 11 months slower to do so compared with countries in the highest GNI quartile. Coverage disparities were also observed in high-risk populations, including healthcare workers and older adults; however, these were smaller than population-wide coverage disparities reflecting the prioritisation of highest risk populations. mRNA vaccines were more likely to be introduced among countries in the highest GNI quartile than those in lower quartiles, while the inverse was true for inactivated vaccines. This variability in vaccine distribution—both in terms of the products used and the timing and scale of their introduction—reflects multiple factors, including bilateral agreements with pharmaceutical companies[22], procurement and donation agreements via COVAX[23], as well as in-country logistical advantages and disadvantages of different product (e.g. cold versus ultra-cold storage[24,25]). The literature on primary series VE is dominated by data on mRNA vaccines from countries in the highest GNI quartile, with data from lower-income settings and for non-mRNA vaccines being fewer and appearing later. The time difference between the first publication of VE data in the highest and lowest GNI quartiles was over 9 months.

The inequities in vaccine introduction and scale-up left LMICs vulnerable to higher rates of COVID-19 transmission and mortality, and all countries at increased risk of new waves caused by emerging strains[26]. The political, economic, and logistical factors underlying these inequities are highly multifaceted, with each country charting a different course through the process of negotiation and implementation. Key factors contributing to the broad trends quantified here include: the concentration of vaccine manufacturing capacity in high-income countries; monopolisation of vaccine supplies via bilateral purchasing agreements between high-income countries and vaccine manufacturers[22,27]; and the recognition by late 2021 that booster doses were necessary to reinforce protection in the context of waning immunity and emerging variants, leading high-income countries to

retain available doses for booster campaigns rather than distribute them to LMICs for primary vaccination[28]. Against this backdrop, the pandemic prompted unprecedented efforts to promote the equitable distribution of COVID-19 vaccines. COVAX, the vaccine pillar of ACT-A (Access to COVID-19 Tools Accelerator), was launched in April 2020 as a multi-partner collaboration to promote the development, manufacture, fair allocation, and programmatic delivery of COVID-19 vaccines. By the time COVAX drew to a close in December 2023, it had raised over $12 billion USD in funding and shipped approximately 2 billion vaccine doses to 146 economies through its supply entity known as the COVAX Facility[29]. However, vaccine supply to LMICs via COVAX did not reach large and predictable volumes until December 2021, hindering large-scale rollout before 2022. In GNI quartiles 1 and 2, the earliest countries to reach the milestone of 40% population coverage were Mongolia, Bhutan, Cambodia, and Ecuador. As detailed further in country-specific commentaries[30–33], swift scale-up in these countries was associated with early purchasing of vaccines outside of the COVAX facility (including from manufacturers based in China and India), strong existing immunisation infrastructure, and early investment to expand implementation capacity.

The only metric in which we did not observe clear global disparities was VE itself. Across three products eligible for inclusion, we observed broadly similar pooled VE estimates across GNI quartiles, with no significant differences in protection against severe COVID-19 according to income status. The absence of clear VE differences across GNI quartiles was consistent for Delta and Omicron variants, albeit with lower VE estimates for the latter in all GNI quartiles—consistent with evidence of immune escape and consequent reduction in protection[34,35]. Our findings are consistent with phase 3 trial data for BNT162b2, in which short-term efficacy estimates against symptomatic COVID-19 were similar in the USA, Argentina, and Brazil[36]. Early efficacy data for ChAdOx1-S were also similar in the UK and Brazil[37]. However, we observed high within-country variability in several variant-specific pooled VE estimates. This likely reflects differences in populations studied, follow-up duration, and outcome definitions, among others. These findings highlight the need for tailored analyses, such as stratified VE estimates within key subpopulations, to better understand the factors shaping VE within countries. No inactivated vaccines met our meta-regression inclusion criteria. Despite reduced immunogenicity and lower VE estimates compared with other platforms, substantial protection against severe COVID-19 has been documented with inactivated vaccines across multiple income settings[16], including during the Omicron period[38,39]. Together, these findings suggest that the

protection offered by COVID-19 vaccines against severe disease is generalisable across populations with diverse demographic and clinical characteristics.

Previous studies have highlighted an unequal distribution of COVID-19 research[40–42]. However, the linkage of a comprehensive public database of VE estimates to country-level income status is a key strength of our study, and enabled us to quantify disparities in the timing and platform distribution of VE data availability. VE studies were undertaken earlier and were more numerous in countries with: early access to vaccines; resources to vaccinate populations at speed; robust health data infrastructure; high testing rates for SARS-CoV-2; and VE study platforms and established methodological expertise for other pathogens like influenza[43,44] (e.g. Israel[45], UK[1], USA[46], and Denmark[47]). Thus, the delayed introduction and scale-up of vaccines in LMICs—as quantified here—served as just one of multiple bottlenecks for COVID-19 VE evaluation, undermining the delivery of timely real-world studies. The obstacles facing COVID-19 VE studies in LMICs sit within a broader array of structural barriers that limit or delay the conduct and publication of relevant research from LMICs, including inadequate support for developing and retaining human resources, the financial burden of conducting studies, and language barriers[48–50].

Real-world VE assesses the effect of specific vaccines against specific outcomes in a given context. Studies across multiple income settings are crucial to comprehensively document the potential range of vaccine performance, considering variation in factors such as age and social structure of the population, disease prevalence, healthcare access, frequency of chronic illnesses, and timely and sufficient vaccine procurement. However, as detailed in this paper, limited and delayed evidence was available on vaccines predominantly used in LMICs[41,51]. The gaps in VE data have presented a significant challenge to policy-makers—key decisions relating to the use of COVID-19 vaccines in LMICs have required the extrapolation of findings from mRNA vaccines being administered in high-income countries. The paucity of context-specific evidence may, in turn, have undermined confidence in and global uptake of COVID-19 vaccines.

Notably, multiple organisations proactively supported the conduct of VE studies in LMICs, including the WHO, CEPI, and the Bill and Melinda Gates Foundation. For instance, four WHO regional offices (American, African, European, and Eastern Mediterranean) supported the creation of COVID-19 VE networks to share protocols, to build country-level capacity for conducting these studies, to have a forum for sharing challenges and successes, and to pool results[52–55]. Delays in vaccination and in identifying funding prevented launching many of these VE studies early. When studies were launched, low incidence of COVID-19, reduced testing rates, and low vaccine coverage at study sites impacted the ability to obtain the minimum sample size to estimate VE[56]. Lessons learned from trying to set up those VE studies, such as the efficiency and speed of leveraging existing respiratory disease surveillance platforms, should be applied to future pandemic preparedness and vaccine introduction planning to ensure that policy decisions are informed by VE estimates from diverse settings.

Strengths of our study include the linkages made across large public datasets and the use of a harmonised approach to quantify disparities in COVID-19 vaccine implementation and VE research. The VIEW-hub database includes studies identified via 11 repositories and is therefore likely to offer a comprehensive and representative database of preprint and published literature. We adopted a multi-level meta-regression approach to address the nested structure of the available VE data and observed consistent findings with respect to country income status across vaccine products and variants[57,58].

As a secondary analysis, we are subject to the biases and limitations of the data sources, such as residual confounding within VE studies, and potential variation in the completeness, timeliness, and reliability of vaccine coverage estimates[4]. Criteria used to define healthcare workers were not available and age thresholds to define

older adults were frequently missing. The high coverage in these populations (e.g. 100% median coverage among healthcare workers in GNI quartile 2 as of December 2023) may reflect an underestimation of the target denominator (defined by International Labor Organization statistics) relative to the coverage numerators captured by WHO reporting systems. Our meta-regression was underpowered to detect geographic differences in VE due to the limited data available for lower-income quartiles, and there were not enough studies to perform sensitivity analyses exploring potential confounders and effect modifiers, such as age.

Several other limitations of the study should also be acknowledged. First, we did not consider variation in the implementation or VE of booster doses, which may have exhibited different patterns compared with primary series vaccination, especially in the context of increasing hybrid immunity, which might have varied across country income strata[59,60]. Second, given that our aim was to leverage public datasets, a risk of bias analysis was beyond the scope of our study. Several previous reviews have highlighted frequent moderate or severe risk of bias in COVID-19 VE studies[61,62], often related to incomplete adjustment for confounders (e.g. clinical or socioeconomic status). Finally, due to the impact of waning immunity on VE, we report on variation in the maximum potential follow-up time across studies, although follow-up may have been shorter for many individuals enroled in each study. However, our findings were consistent in a sensitivity analysis restricted to the first 26 weeks after primary vaccination.

Notwithstanding the speed and breadth of the global rollout of COVID-19 vaccination, the rush to secure vaccines and protect populations led to stark inequities in implementation. In LMICs, earlier availability of vaccines and higher uptake could have averted a significant proportion of COVID-19 morbidity and mortality, along with associated economic and health system impacts. Disparities in VE research have created key evidence gaps for policy-makers and hindered context-specific optimisation of vaccine programmes. The inequities in vaccine implementation and VE research observed for COVID-19 serve as a benchmark against which to measure progress in future pandemics. In the fall-out of COVID-19, we must lay the foundations to do better next time[28,63].

## Methods
### Data sources
All data used in this study are publicly available, aggregated by country/territory, and contain no individual-level information. Vaccine introduction dates were obtained from the World Health Organization (WHO) COVID-19 database and the Our World in Data (OWID) COVID-19 vaccination database[4], both as of 07 January 2024. These resources pool national vaccination data from various sources, including reports and dashboards maintained by governments. Primary series vaccine coverage data were obtained from the OWID COVID-19 vaccination database. The definition of 'primary series' is specified by the product-specific use authorisation and can vary by product and country/territory. In most cases, this represents two doses, although the Ad26.COV2.S vaccine (widely implemented as a single dose primary series) is a notable exception. Country-/territory-level income status was obtained from World Bank estimates of per capita Gross National Income (GNI) based on the Atlas method[64]. We selected data for 2019 (the most recent pre-pandemic data point) where available[65]. If 2019 data were unavailable for a country/territory, we used the most recent data prior to this date. The Marshall Islands, the Federated States of Micronesia, and Puerto Rico were excluded from descriptive analyses as vaccine coverage metrics were not included in WHO or OWID databases despite records of vaccine introductions in the public domain.

Primary series coverage estimates for high-risk populations (healthcare workers and older adults) and overall population coverage

estimates as of December 2021 and December 2023 (marking the end of COVAX) were obtained from the WHO/UNICEF COVID-19 Vaccination Information Hub[12]. This collates health workforce data from the International Labor Organization, older adult and overall population denominators using data from the United Nations Population Division, and immunisation data from the WHO/UNICEF electronic Joint Reporting Form (eJRF), alongside input from other regional reporting systems. Healthcare worker definitions may vary by country/territory and were not included in the database. Age thresholds to define older adults were available for a subset of countries/territories, as summarised in Supplementary Table 3, and typically ranged between 50–65 years. Completeness of coverage data varied by GNI quartile, as summarised in Supplementary Table 2. Data for November 2023 were used if data for December 2023 were unavailable.

VE studies from published and preprint sources were obtained on 11 January 2024 from the VIEW-hub living literature review conducted by the IVAC at the Johns Hopkins School of Public Health, with support from CEPI and WHO[66]. Briefly, the VIEW-hub database captures COVID-19 VE studies from preprint and published literature. Only studies with sufficient methodological rigour and detail are included. VE estimates are required to include confidence intervals and account for confounding variables. Primary series studies must involve individuals with and without the clinical outcome of interest. COVID-19 laboratory-confirmed outcomes and documented vaccination status for most participants are also required. Other inclusion and exclusion criteria can be found on the VIEW-hub website[13]. If individual studies in VIEW-hub reported multiple country-/territory-specific VE estimates, we assigned these estimates to separate study IDs.

## Descriptive analysis

Countries and territories were categorised by income status according to GNI quartile. We then compared several metrics of vaccine implementation and VE evaluation by GNI quartile. The date of COVID-19 vaccine introduction in each country/territory was defined by taking the earliest date from the WHO and OWID databases (thereby leveraging the complementary data flows of these two resources). To assess disparities in subsequent scale-up of the vaccine programme, we quantified the proportion of countries/territories in each GNI quartile achieving 40% primary series coverage by the end of 2021 and 70% primary series coverage by mid-2022—milestones aligning with the WHO global COVID-19 vaccination strategy[10]. We also quantified the time in days between vaccine introduction and the achievement of these coverage milestones. VE data availability was determined based on the first date of publication of country-/territory-specific primary series VE data, allowing outcomes of any severity. We used the earliest preprint date where applicable for analyses relating to the timing of evidence availability (i.e. taking the publication date of the first preprint if this was superseded by later versions). As a secondary analysis, we used Spearman's rank correlation coefficient to assess the association between GNI as a continuous variable and: vaccine introduction, vaccine scale-up, and first VE publication.

## Meta-regression of VE by country/territory income status

Among the VE estimates included in our descriptive analysis, we applied further inclusion and exclusion criteria based on VIEW-hub metadata to ensure a minimum degree of comparability required for meta-regression. The outcomes of interest were severe COVID-19 and COVID-19-related mortality. We selected variant-specific VE estimates relating to adult populations (majority of included age range ≥18 years), complete homologous primary series vaccination, and cohort or case-control study designs. We excluded VE estimates focusing exclusively on specific populations (e.g. healthcare workers, immunocompromised individuals) or individuals with prior infections. We included estimates relating to complete primary vaccine series (generally defined as 1 dose for Ad26.COV2.S and 2 doses for other products). We excluded vaccine products and variants with fewer than 10 unique studies. We stratified analyses by product, outcome severity, and dominant circulating variant.

To proceed with model fitting, we required sufficient representation across multiple GNI quartiles (at least 10 studies, with 2 or more falling outside the majority quartile). To prevent over-representation[67], if studies presented separate VE estimates for the full follow-up period alongside nested sub-periods, we excluded the nested sub-periods from the analysis. If studies reported separate VE estimates only for sequential sub-periods, we excluded later periods to mitigate the effect of waning. If studies reported VE estimates for multiple Omicron sublineages combined and also separately for each nested sublineage, we excluded the nested sublineage estimates. If studies reported separate VE estimates for COVID-19-related hospitalisation and severe COVID-19, we retained hospitalisation estimates to enhance comparability across studies. If only stratified VE estimates were reported for sublineages (e.g. Omicron BA.1 and Omicron BA.2) or age (e.g. 60–69 and 70–79 years of age), the estimates were included separately, though country-/territory-level clustering was accounted for in the statistical models. For VE estimates relating to complete follow-up periods of a given study, we defined the upper limit of follow-up (maximum potential duration of follow-up in weeks) based on the 'max duration follow-up' variable in VIEW-hub. This represents the maximum potential elapsed time from 14 days after primary vaccination to the end of the study period and is available for studies published from late May 2021. For VE estimates relating to specific sub-periods (e.g. 2–24 weeks within a longer study), we defined the upper limit of follow-up (in weeks) based on the end of the sub-period.

Data processing, statistical analysis, and visualisations were performed in R version 4.4.1 (2024-06-14) using the *metafor* (version 4.6-0) and *dmetar* (version 0.1.0) packages[58]. The primary covariate of interest was GNI quartile. We fitted three-level meta-analytic models to obtain pooled VE estimates, incorporating GNI quartile as a moderator variable. Three-level meta-analyses enable nested structures (clusters) in data to be explicitly modelled[58,67]. We anticipated clustering of individual VE estimates by country/territory, given country-/territory-level variation in vaccine programme eligibility and implementation, the timing of waves, and healthcare infrastructure. Weights were assigned to each estimate based on their standard errors. As described in the Supplementary Methods, VE and confidence intervals were transformed into log risk ratios for meta-regression models[57,58,68]. Restricted maximum likelihood (REML) methods were used to estimate variance components representing heterogeneity within countries/territories ($\tau^2_{Level2}$) and between countries/territories ($\tau^2_{Level3}$). Additionally, we used the $I^2$ heterogeneity statistic to quantify the proportion of the variation in effect sizes attributable to heterogeneity within countries/territories ($I^2$ level 2) and between countries/territories ($I^2$ level 3)[58]. Pooled risk ratios for each GNI quartile were calculated based on the three-level models, then converted to VE estimates by taking [(1- pooled risk ratio)*100]. A moderation test was used to assess whether differences between quartiles were statistically significant. Prediction intervals were also determined for each GNI quartile estimate[58]. GNI quartile was coded as a categorical variable with quartile 4 (highest income) as the reference. We conducted sensitivity analyses restricted to (i) VE estimates for individuals over 60 years of age and (ii) VE estimates with a maximum upper limit of follow-up of 26 weeks, although only the latter yielded comparisons meeting our eligibility criteria for meta-regression (Supplementary Table 4).

## Reporting summary

Further information on research design is available in the Nature Portfolio Reporting Summary linked to this article.

## Data availability

Raw and processed data are available on GitHub (https://github.com/marrpesce/COVID-global-vaccine-disparities; https://doi.org/10.5281/zenodo.16909975). Curated datasets generated in this study are provided in Supplementary Data files 1 and 2.

## Code availability

All data and analysis code related to this study are available on GitHub (https://github.com/marrpesce/COVID-global-vaccine-disparities; https://doi.org/10.5281/zenodo.16909975). To enhance reproducibility, the R package *renv* (version 1.1.4) was used to capture specific package versions required for the analysis.

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

## Acknowledgements

M.P. received support from a Chevening Scholarship, funded by the UK government's Foreign, Commonwealth and Development Office and partner organisations. E.P.K.P. is funded by the National Institute for Health and Care Research (NIHR) Health Protection Research Unit in Vaccines and Immunisation (NIHR200929), a partnership between UK Health Security Agency and the London School of Hygiene and Tropical Medicine. The views expressed are those of the author(s) and not necessarily those of the NIHR, UK Health Security Agency, or the Department of Health and Social Care.

## Author contributions

Conceptualisation: M.P., E.P.K.P.; Data curation: M.P., E.P.K.P. and M.M.H.; Formal Analysis: M.P., E.P.K.P.; Methodology: M.P., E.P.K.P., M.M.H., M.K.P. and D.R.F.; Project administration: M.P., E.P.K.P.; Software: M.P., E.P.K.P.; Visualisation: M.P., E.P.K.P.; Writing—original draft: M.P., E.P.K.P.; Writing—review & editing: D.R.F., M.M.H., K.L.O'.B., M.K.P., A.R., C.V., A.W.-S. and E.P.K.P.

## Competing interests

The authors declare no competing interests. D.R.F., K.L.O'.B., M.K.P., and A.W.-S. are WHO employees. The views, findings, and conclusions in this report are those of the authors and do not necessarily represent those of the World Health Organization.
