## [Peer Review file · Nature Communications]

Global disparities in the introduction, scale-up, and evaluation of the effectiveness of COVID-19 vaccines

Corresponding Author: Dr Edward Parker

Version 0:

Reviewer comments:

Reviewer #1

(Remarks to the Author)

We thank the authors for an interesting study examining global disparities in vaccine uptake, rollout and effectiveness. While our feedback points are mostly minor (see below), there are a few major points to consider.

Major:

- More consideration may need to be given to variation in countries' dosing schedule. For example, it appears that countries are compared "like-for-like" even though vaccination schedules differ both in number and timing of doses. What consequences will this have for the results on VE.
- What new insights does the GNI analysis provide beyond the known disparities caused by wealthier countries' access to Covid-19 vaccines.
- Clarify expectations regarding why VE, measured by severe Covid-19 outcomes, would differ by country. Variations may be due to timing of VE studies in relation to dominant variants, but also by each countries Covid-19 vaccination policies.

Minor (section-by-section)

Abstract:

- The abstract would benefit from greater clarity. For instance, differentiate between introductory information and research findings. The phrase "Covid-19 vaccines were the most rapid" is incomplete—rapid what?
- Rather than stating that vaccines were implemented across "all geographies," consider being more specific.
- Are the findings genuinely novel? If not, highlight what's new about this analysis. Have no other studies on VE in low-income settings been conducted?

Main Text:

Global Disparities in Vaccine Introduction and Scale-Up:

- Provide a definition for GNI.
- Does the study need an ethical statement clarifying that all study data is publicly available, aggregated by country, and non-identifiable?
- In Lines 33 to 39, the sentences seem contradictory: the first states that there's robust protection and no significant differences in VE at the country level, while the next sentence mentions stark disparities across the globe.
- In Lines 41 to 45, the two sentences also appear contradictory. Instead of stating that vaccines were widely implemented in December 2020 and then highlighting global disparities, perhaps provide context. For example, mention that vaccines were distributed widely in high-income countries (HICs) while low- and middle-income countries (LMICs) lagged behind. Cite examples of later Covid-19 vaccine implementation in LMICs.
 - o Alternatively, start with Line 45 and link assessments of VE with the vaccine rollout, noting the discrepancies between HICs and LMICs.
- "Minimum vaccination date" doesn't make sense. Did you mean "earliest vaccination date"?
- The sentence "had recorded vaccinations" could be improved for readability. Perhaps rephrase it as "had recorded at least one vaccination." Additionally, the median time to vaccine introduction might be clearer if stated as: "The median time to vaccine introduction was 20 days (IQR 15-41) in countries with the highest GNI quartile, 57 days (21-77) in the second highest, and 78 (53-95) and 103 (91-124) in the second lowest and lowest quartiles, respectively."
- In Line 45, a smoother transition between the first two sentences on vaccine implementation and effectiveness is needed.

This could be achieved by linking the ideas earlier, so the transition isn't as abrupt.

- The mean time reporting in Line 65 is confusing with the different days and numbers in parentheses (20, 57, 78, 103). These need better labeling or clearer explanation.
- Justify the choice of approval date in Line 63.
 - o For example, while December 8th marks the first vaccine administered in the UK, note that vaccines were approved earlier.
- What does "moderate consistency" mean? Clarify by specifying the agreement between WHO and OWID databases. For example, explain which data source was used for the median time estimates and how alignment with the other database affected the results.
- Define "primary series" clearly.
- The milestone mentioned seems like an afterthought; provide more details about it.
- Identify which low-income country performed well and whether there are lessons to be learned from it.
- Clarify the meaning of "without date restriction" and ensure the numbers are presented relative to GNI groups.
- Lines 63 to 70 could be presented more clearly.
 - o Lines 80 to 84 have a similar issue. Reporting styles like "respectively" can disrupt readability, and the data might be better displayed in a table.
- In Line 92, the phrase "In the context of limited supply" seems circular. Rephrase to clarify the impact on high-risk groups.
- In Lines 92 to 100, explain whether the call to vaccinate high-risk groups (e.g., the elderly and healthcare workers) was consistent across countries. The discrepancy between 10% vaccination coverage for older persons and 56% for healthcare workers in the lowest quartile needs more context.
 - o Is the median of 100% coverage for healthcare workers in the second-lowest income group correct? Could there be biases in the data?
 - o How are "older adults" defined in each country? Include this in the supplement if necessary.
- Clarify what specific differences were smaller, as it's not clear which data comparisons are being made.

Global Differences in the Evaluation of Vaccine Effectiveness (VE):

- Provide a clear definition of VE used in the study.
- In Line 121, when introducing the section on global differences in VE evaluation, explain what VIEW-hub is and how you selected the studies.
- Line 127 is missing "respectively" or a similar word for clarity.
- Has a comprehensive review of VE studies been conducted before? Clarify if this is new.
- In Line 133, change "quartiles" to "GNI quartiles."
- When you mention that a country had a VE study, clarify whether this means the study examined VE in the country's population or whether the authors were from that country.
- In the paragraph starting in Line 148, you seem to imply there are multiple VE metrics. Clarify the differences between these metrics.
 - Provide more details on why VE might differ due to demographics, comorbidities, or social mixing for each VE measure. Non-pharmaceutical interventions (NPIs) should also be discussed.
- Lines 163 to 165 seem like a major point. Consider moving it to the discussion and expanding on it there.
- In Line 171, "meta-regression" is introduced too late. Prime the reader earlier in the report.
- How were populations within studies accounted for? If VE varies due to characteristics like NPIs and contact patterns, wouldn't VE estimates also vary widely within populations?
- Explain how pooled VE estimates were calculated, particularly considering different study population sizes.
- Given that VE wanes over time, how might the timing of studies impact VE calculations? Add this as a limitation if necessary.
- In Line 180, clarify what is meant by "upper-limit of follow-up."
- Line 182 reports no significant differences in VE but also mentions variations. Can these variations be explored in greater depth?
 - In Line 184, what is I^2 ? Provide an explanation.
- The definition of VE is confusing in the context of both severe Covid-19 and Covid-19 mortality. Wouldn't these require two different VE metrics? How are they combined?
- Remove the summary in Lines 206 to 208 and begin with a discussion instead.
- In Line 283, add a period between "research" and "The VIEW-hub."
- Lines 282 to 287 should introduce the "Global differences in..." section more clearly.
- Remove the introductory sentence in Lines 307 to 310 and start with a statement on vaccine rollout disparities and the implications of inequities in implementation.
- Line 495: Include a disclaimer regarding any exclusion criteria used in the VE studies and provide more detail on this.
- Grouping GNI quartiles might result in arbitrary classifications. Discuss the limitations of using this method.
- A table listing all VE estimates by study would improve reproducibility and indicate which estimates were used or excluded.
- Could a country's vaccine dosing schedule (e.g., two doses three months apart versus two doses six months apart) impact VE? Discuss this.

Discussion:

- Strengthen the discussion by expanding on the significance of findings, such as the disparity in VE estimates across income groups. Address how these disparities impact future vaccine rollouts and effectiveness evaluations.
- In Lines 213 and 216, discuss the role of mRNA vaccine costs in wealthier countries and how this may have contributed to disparities. Expand on the variation in coverage disparity and its implications.
- In Line 241, relate comments about VE in HICs to studies and consider adding a citation.

- In Line 291, could sensitivity analyses be performed in data-rich settings like GNI3/4?
- The concluding section should address the "what's next" question. How does this research contribute to future studies or policy actions?

Methods:

- Highlight how the study's methodological decisions align with Cochrane guidelines (Line 491). In the file "WeeklySummary_COVID19_VE_Studies_20240111.xlsx" in the "Primary Series Studies" tab, the variables "dose_number" and "dose" are unclear. How do they fit into the definition of the primary series? If VE estimates are mixed between one or two doses, how will this impact the study's results?

Figures:

- Figure 1: Consider reorganizing the figure for better clarity and resolution. Using a full-page layout or separating the subfigures could improve readability. Specifically, breaking the maps (A) into their own figure and grouping other components separately would help. The current resolution is low, and small world maps are hard to interpret.
 - o In 1A, the grey shading for "not reached" is difficult to distinguish. Darker greys (e.g., for Greenland and Syria) are more visible.
 - o In 1B, with ~195 countries divided into quintiles, can the countries in each quintile be listed somewhere (e.g., in a supplement)?
- Figure 2: Clarify whether you mean the "whole population" or the target group within each country, which can vary. If a country expanded vaccination to children under 18, the coverage rate could be skewed. Make sure the numerators and denominators are meaningful for cross-country comparisons.
- Figure 3: This figure also needs to be broken down. Presenting subfigure (A) separately and grouping the others logically would enhance clarity.
- Figure 4: Increase clarity by enlarging the figure and possibly organizing it in a grid. Remind readers why no data is shown for GNI quartile 1. Could the lower VE values in GNI quartile 2 be due to study timing during the Omicron wave, which lowered VE?

(Remarks on code availability)

The code was scanned over and seemed appropriate. However, the data used for the analyses seems to compare countries "like-for-like" when in fact differences in the definition of primary series seems to occur (i.e. 1 v 2 doses).

Reviewer #2

(Remarks to the Author)

(Remarks on code availability)

Reviewer #3

(Remarks to the Author)

Summary

Our read of the work is it intends to highlight the correlation between national economic performance and vaccine program performance. The work concludes that higher gross national income per capita (GNI) correlates with better vaccine emergency response program measures: earlier initial doses, earlier achievement of coverage milestones, and earlier post-distribution evaluation of the response technology. This work also includes a meta-analysis of vaccine efficacy (VE) to assess if that varies with GNI, and concludes that it does not. Were such claims clearly proven and well-quantified, that would be useful information for future public health planning efforts, and thus significant.

However, we are concerned with overall methodology and especially the analytical approach to the data. We think these concerns undermine the potential usefulness of this work, but that those concerns can potentially be remedied. As such, we have several suggested improvements. Briefly, we think this work would be improved by:

- Presenting correlation plots and calculations rather than quartile visualizations. Linear and rank regression plots would be clearer about the overall quality of correlation between GNI and the various vaccine metrics.
- Dropping the meta-analysis of VE. This muddles the apparent research question, which seems to focus on program performance. We agree with assessing the timing and completion of evaluation work as a useful indicator of the overall emergency response program, but the VE comparison seems to be distinct work and likely requires different statistical machinery to do properly.
- Clarifying the status of evidence for various conclusions, both as found in this and other work, and in particular what measures had a demonstrated casual chain versus observed correlations.

Lastly, we are reading this work as researchers that focus on infectious disease modeling. We could imagine using this kind

of work as, say, the basis for selecting archetypal settings in a scenario analysis. The calculations and visualizations presented in this paper could be useful for understanding the observed variation in the data, but of course would not necessarily match those hypothetical analytic needs (e.g. we might be interested in 3 groups instead of 4). For our kind of use case, the most valuable result would be a well curated, documented, and organized data set; that seems to be roughly present, but in need of refinement and highlighting.

Below, we expand on the summary points above, and conclude with some detailed notes.

Research Questions & Findings

Our interpretation of the work is that it intends to evaluate the correlation between economic performance (as indicated by GNI) and vaccine programme performance (as indicated by timing of initiation, of achieving key coverage milestones, and of evaluating programme benefits). The work also intends to investigate correlation between GNI and assessed VE.

The work concludes that GNI and various programme performance indicators are correlated. This is argued by visualizing the difference in vaccine programme distributions between GNI quartiles. The work also concludes that GNI and VE are not correlated, by illustrating that the pooled VE estimates have overlapping uncertainty intervals, and checking that covariates are insignificant.

How did we ask those questions and what results were found?

The analysis combined a variety of country-level data sources on income and vaccine program implementation. GNI per capita for 195 countries was obtained from the World Bank. The date of vaccine introduction was taken as the earliest date listed in the WHO and OWID databases and was available for 194 countries. Data on primary series vaccine coverage was obtained from the OWID database for all but three countries/territories.

The analysis compares 1) timing of vaccine introduction, 2) scale up of vaccination, and 3) evaluation of VE. The metric used to assess introduction timing was minimum recorded vaccination date. The metrics used to assess scale-up were 1) whether a country achieved 40% coverage by December 31, 2021, 2) time to achieve 40% coverage, and 3) coverage (stratified by risk group) as of December 2021 and 2023.

The metrics used to assess VE evaluation were 1) number of studies available, 2) time from introduction until first study publication, and 3) percentage of studies stratified by vaccine platform. To perform this evaluation, the analysis considers published, preprint and gray literature sources from the VIEW-hub literature review from Johns Hopkins IVAC. Studies with multi-country VE results and/or those from countries which lacked available GNI data were eliminated, resulting in 422 (out of 434) studies used for the authors' descriptive analysis. For example, studies from Cuba and Taiwan were eliminated because World Bank GNI data was not available for these countries. Overall, VE data was available for 47 out of the 192 countries reporting vaccine introduction.

Differences across countries were observed by reporting these metrics by GNI quartile and by plotting boxplots of each metric stratified by GNI quartile. Additionally choropleths were utilized to show variations in these metrics by country. No statistical tests were performed to quantify the significance of these differences.

A meta-regression was performed with the intent to assess differences in VE by GNI. Analysis was stratified by vaccine product and variant. 53 out of the 422 studies used for the descriptive analysis were selected for this meta-regression based on criteria such as virus variant, outcome of infection, and number of studies by GNI quartile.

Should we believe it?

The current analysis relies on looking at differences across income quartiles to demonstrate the supposed association in vaccine program performance. However, the categorization into quartiles is somewhat arbitrary and does not fully use the data. Presentation of correlation plots and calculations in combination with regression calculations - that is, computing linear and/or rank correlation coefficients without stratifying into quartiles - would provide a more convincing argument and better use all the available data.

The analysis of vaccine efficacy seems distinct from the other research questions being addressed in this paper. The relevant question seems to be "are these GNI quartiles different?", but the meta-regression analysis does not seem suited to answer it. While not framed this way in the manuscript, we read the conclusion here as "we cannot clearly see if there is a difference". The analysis concludes that GNI as a meta-regression covariate is insignificant, but the more appropriate statistical comparison here would be "is the difference between strata clearly less than X%?". We expect the likely answer to that would be "no" for large enough X so as to be uninteresting, given how wide the measurement uncertainty intervals are here. However: we do not think this part of the analysis serves what we read as the overall aim of the work. We think the aim here is to understand relative operational pace in vaccine programs, which of course includes post-distribution elements like ongoing evaluation; the actual VEs concluded by those ongoing evaluation programs is a separate issue. A more relevant issue for VE studies would be something like whether studies meet minimal quality standards (which publication is not necessarily an indicator of).

Lastly, much of the language in the discussion and conclusion sections implies causal relationships. That language is generally not supported by the specific analyses here, nor by citations, and generally falls into the "everybody knows X" category. This is a framing error, which we find undermines the actual analysis and results that could be presented here.

Detailed Notes

Typos/grammar

Line 143: Missing closing parenthesis

Line 25: Stray “)”

Line 283: Missing period after “VE research”.

Line 448: Extended Data Fig. 12 is referenced, but only 8 extended data figures are presented.

Line 452: “Primary series coverage estimates of high-risk populations...”

Lines 518 - 526: Font size is smaller than the rest of the text.

Figures

In Figure 1A it is difficult to distinguish between “not reached” and “90+”.

In Figure 2, shifts in coverage from December 2021 to December 2023 are impossible to distinguish given the layout - perhaps it would be better to show side-by-side boxplots of both dates within each quartile (although see broader comments about alternatives to boxplots).

In general, the color distinction choices do not seem color-blind friendly.

References needed

There are assertions made in the discussion / conclusions that would benefit from more thorough citations. We list a few examples below.

Lines 217-22: “This variability in vaccine distribution...”

This statement hints at causality, the implication being that variation in income drives variation in the logistics/agreements which in turn explains the observed differences in this analysis. The analysis presented does not include any information on country-level logistics or vaccine-related agreements, so citation of work demonstrating these differences by income and their downstream impacts is needed.

Lines 242-244: “...robust health data infrastructure; high testing rates for SARS-CoV-2; and VE study platforms and established methodological expertise for other pathogens like influenza.”

The current analysis describes differences in timing and numeracy of VE studies by income, however country-level resources, infrastructure, and pre-existing expertise were not directly assessed. While we share the intuition that these factors vary with income, the authors should cite work conclusively showing this.

Lines 276-278: “When studies were launched...”

Again without citations this seems to be a statement based on intuition. Has any work been done to investigate the barriers to meeting recruitment targets? It seems that some of the barriers mentioned would only be issues for certain study types and not others (e.g. observation cohorts vs. RCTs).

The COVAX program is used to determine the date cutoff for scale-up analyses. It would therefore be prudent to reference the relevant policy information.

Line 141: Include citation for the paper from Zambia within the text.

Other clarifications/comments

The quartiles division of GNI is a bit arbitrary. Why not use something like Spearman correlation on date rank vs GNI rank, or Pearson correlation on GNI value? We feel that presenting correlation plots as well as linear and/or rank regression plots would provide a clearer picture of the association between GNI and the individual vaccine metrics.

The authors analysis of inequities in vaccine in introduction and scale-up considers the percentage of countries that achieve test coverage/ scale-up by Dec 31, 2021 but then compares the median for time until introduction/ scale-up without a date restriction. A more consistent analysis would be to consider the median time for countries that completed the milestone of scale-up by Dec 31st versus the median time for all countries (including those that did not meet the goal).

For VE studies, how are publication dates interpreted? Are the dates for preprints vs published studies comparable? Likely many of the published studies were also preprints at some point - would it make sense to always use preprint date if available?

All acronyms should be defined upon first use in the main text.

Clarification is needed in the text itself that GNI is on a per capita basis.

Given the use of the end of COVAX as a cutoff point in the analysis, a brief description of COVAX itself should be included.

In general, more discussion of the role that differential age structure may have in driving the observed differences in vaccine coverage across countries is warranted.

On line 61 the authors state that 194 countries had records of COVID-19 vaccine introduction as of 31 December 2021, but on line 138 it states that 192 reported vaccine introduction. Please clarify why those two numbers differ.

The code runs for the most part, although we did not dig too deeply at this stage point. There is an apparent error in script 4

(selecting a column that doesn't exist) and some of the required packages don't work with the most current version of R. The authors may want to use something like `renv` to ensure particular versions.

Overall we find the most valuable result of this work for our use is the curated dataset of measures relevant to vaccine program assessment. We think the authors could take two additional steps to make this even more valuable for readers like us:

Efforts to fill in data gaps from other sources. For example, World Bank GNI data was not available for Cuba, but UN data is available. In the several places where there are a small number of missing entries, the authors could improve the resulting dataset by synthesizing more sources.

Provision of a clean data table which for each country shows GNI and all the metrics of vaccine program performance used in this analysis. It seems that `lneq_merged_with_vaccine_data_plus_priority_groups.csv` comes closest to this, but it is not well documented. It would be more helpful to have a more user-friendly version (easier to find, more informative column names / fewer extraneous columns, a data dictionary / schema definition, etc.) highlighted as a key output of this analysis.

(Remarks on code availability)

We attempted to run the code. At this stage, our review was cursory. We did not find that the code "just worked", but it seemed like it would with a bit of troubleshooting.

Reviewing the code itself: it generally seemed organized and clearly written. We did not check that the code is a precisely faithful translation of the described methods in the manuscript at this stage.

Reviewer #4

(Remarks to the Author)

(Remarks on code availability)

Reviewer #5

(Remarks to the Author)

(Remarks on code availability)

Version 1:

Reviewer comments:

Reviewer #1

(Remarks to the Author)

The authors have provided a thorough and (more than) satisfactory response to all the issues raised.

(Remarks on code availability)

Reviewer #3

(Remarks to the Author)

Summary

We thank the authors for their detailed consideration of our comments and for addressing the issues that were brought up; these changes have clarified the authors' arguments and strengthened the evidence for those arguments. In particular, the correlation analyses and visualizations using the data without stratification make the trends clear and portable to other analyses. The curated datasets are well-organized, and the README file provided in the GitHub repository makes the analysis workflow clear.

We also find the updated framing of the VE meta-regression analysis clearer and better justified. However, we still have two concerns with this element of the work.

First: overall, there remains language implying the lack of significance provides meaningful insight, e.g. “The only metric in which we did not observe clear global disparities was VE itself.” is ambiguously readable as “Global disparity in VE is small.” - but that is not the statistical question in the analysis or a logical implication of the non-significance in results. It seems to us that the specific analysis here should be more like confirming that the GNI factor contribution to estimate differences is clearly (i.e. statistically significantly) smaller than some threshold X (which the authors can choose, arguing what seems like a reasonable practical value for worrying about HIC vs LIC differences). We are not meta-regression experts, so cannot recommend specific changes to the analysis here if necessary; if these results somehow already show this conclusion (i.e. the effect of GNI category is identifiably smaller than X threshold at an appropriate significance level), the authors should reframe it as such. If such re-analysis is not possible, then changing the framing to be “the analysis cannot tell if GNI has an effect on VE” seems to be the only recourse.

Second, in terms of the technical execution of the meta-regression, we are a bit unclear on precisely what the regression model is. The authors’ use of “three-level model” suggests a standard model, but for lay scientist readers not routinely consuming meta-regression, it would be enormously beneficial to see the regression equation(s) to clarify what the levels are. As part trying to figure out what the levels are, we dug into the code - it appears that the model includes a random effect for the country and an observation-level random effect (defined as `subgroup = as.factor(1:nrow(.))`). Is there supposed to be an observation-level random effect? Why not, say, a study-level random effect? This would be more straightforward to assess with some summary context for readers in discussing the methods, such as would be provided by including the specific equations.

We include additional conceptual questions and comments below, but leave it up to the purview of the editor to review these responses and make a final decision.

General clarifications

Line 105: It stands out that the Q2 coverage is so much higher than the rest. Might warrant some discussion.

Lines 197 – 199: The authors present pooled VE estimates for each GNI quartile. However, only 1 study is used to determine this estimate for GNI quartile 2; is it appropriate to describe as a “pooled” estimate?

Line 208: Negative VE in the CI for Q2 seems odd. Does the data genuinely support negative efficacy? Alternatively, is it an analytical artefact which is potentially addressable via different assumptions?

Lines 236-238: “The time difference between the initial publication of the VE data in the highest and lowest GNI quartiles was over 9 months.” – Do the authors mean to say “median” initial publication?

Methods Line 12: Puerto Rico is not a country. Might be helpful to clarify that territories are being treated as their own “country” in the analyses, if this is the case.

References needed

Lines 230-234: “This variability in vaccine distribution (e.g. cold versus ultra-cold storage).” – This statement could still benefit from having a citation.

Figures

Figure 3: It may be helpful to include the correlation coefficient value in the corner of the plot.

Typos/grammar

Affiliation 3 doesn’t correspond to any of the authors.

Line 98: one of the parentheticals includes IQR, the others do not, and earlier similar quantitative results also do not include the IQR annotation; whatever the convention for stating or not stating the confidence interval definition, it should be applied consistently.

Line 100: Make “Fig. 3B” bold.

Line 135: There is a stray comma after em dash.

Line 160: Make “Fig. 3C” bold.

Lines 182-184: “We selected across multiple GNI quartiles.” – Is this sentence grammatically correct?

Line 201-2: “...series vaccination by was 25...” – This sentence also seems grammatically incorrect; there may be a word missing.

Methods Line 91: “package” should be plural.

(Remarks on code availability)

Yes, after some markdown troubleshooting.

Point-by-point response to reviewer comments

We thank the reviewers for their detailed and constructive comments. Our revised paper has benefitted considerably from these thoughtful critiques, as detailed in our point-by-point responses below. The addition of two curated datasets in the supplementary materials is a particularly valuable suggestion that will complement the public GitHub repository.

Reviewer #1 (Remarks to the Author):

We thank the authors for an interesting study examining global disparities in vaccine uptake, rollout and effectiveness. While our feedback points are mostly minor (see below), there are a few major points to consider.

Major:

(1) More consideration may need to be given to variation in countries' dosing schedule. For example, it appears that countries are compared "like-for-like" even though vaccination schedules differ both in number and timing of doses. What consequences will this have for the results on VE.

Response: We thank the reviewer for raising this point. We designed our meta-regression to ensure a minimum degree of comparability among VE estimates included in a given analysis. Notably, this included restriction to complete primary vaccination series, thereby ensuring consistency in the number of doses being compared in a given analysis. To further clarify this, we have added the following sentence to the methods: "We included estimates relating to complete primary vaccine series (generally defined as 1 dose for Ad26.COV2.S and 2 doses for other products)." We have also made additions elsewhere to explicitly define 'primary series' (see response 17 below).

Notably, while checking our analytic pipeline in response to this suggestion, we noticed the inclusion of one 2-dose study of Ad26.COV2.S. After accounting for this, Ad26.COV2.S no longer met our minimum study count for inclusion in the meta-regression (at least 10 comparable studies) and the vaccine has therefore been excluded from the analysis. We thank the reviewer for drawing attention to this issue.

Other relevant criteria to ensure comparability in our meta-regression include the stratification of estimates by variant and outcome severity (both known to substantially impact VE), and the prioritisation of early VE estimates where applicable to mitigate the effect of waning protection on our meta-regression (also accounted for via a sensitivity analysis limited to estimates relating to the first 26 weeks after vaccination).

We agree with the reviewer that variation in dose interval may be an important factor in shaping vaccine immunogenicity and VE. However, recommended intervals for each product were the same across all countries, and while implementation decisions may have varied (e.g. with countries opting to delay second doses to prioritise first dose coverage), we do not believe dosing interval to be a significant confounder in the potential association between country income status and VE.

(2) What new insights does the GNI analysis provide beyond the known disparities caused by wealthier countries' access to Covid-19 vaccines.

Response: We acknowledge the reviewer's point that disparities in COVID-19 vaccine implementation were well known prior to this study. We now address this in the Background section as follows: "The ensuing vaccine rollout was characterised by marked global disparities in scale and timing, with low- and middle-income countries (LMICs) often facing significant delays in vaccine access compared to high-income countries (HICs)^{3,4}."

The novelty of our analysis is the application of a harmonised analysis approach to quantify disparities across different stages of implementation and evaluation. Thus, we show that delays in vaccine introduction in low-income countries were compounded by subsequent delays in scale-up, and we explicitly quantify these different components. The incorporation of VE studies and

estimates through linkage of VIEW-hub data is another key novelty, as disparities in evaluation data have been given less consideration than those in vaccine introduction and coverage. We have adjusted the Background section as follows to highlight these novel aspects of our study:

"In this study, we harnessed novel linkages across large public datasets to systematically quantify global disparities in the implementation and real-world VE evaluation of COVID-19 vaccines. Using a harmonised analysis approach, we considered the association between country-level income status and (i) vaccine introduction; (ii) vaccine scale-up; (iii) the publication of VE studies; and (iv) the VE against severe disease."

Finally, the GNI analysis underpins our meta-regression of geographic discrepancies in VE (as discussed further in response 3 below).

(3) Clarify expectations regarding why VE, measured by severe Covid-19 outcomes, would differ by country. Variations may be due to timing of VE studies in relation to dominant variants, but also by each countries Covid-19 vaccination policies.

Response: As suggested, we now introduce the rationale for this aspect of the study as follows: *"Protection may also vary according to demographic and clinical characteristics of the vaccinated population, and reduced immune responses and efficacy have been reported in LMICs relative to HICs for multiple vaccines⁵⁻⁸. This may limit the generalisability of VE findings across geographically disparate settings."*

See response 1 above on our criteria to ensure a minimum degree of comparability among VE estimates during meta-regression, including: (1) restriction to primary vaccination (i.e., 2 doses for the vaccines included in the final analysis); (2) stratification by product; (3) stratification by variant; and (4) prioritisation of early VE estimate to account for waning protection.

Minor (section-by-section)

Abstract:

(4) The abstract would benefit from greater clarity. For instance, differentiate between introductory information and research findings. The phrase "Covid-19 vaccines were the most rapid" is incomplete – rapid what?

Response: Thanks for this comment. As suggested, we have separated introductory information and methods from research findings via the phrase *"Our results show that..."*.

The first sentence has been amended to improve clarity, as follows: *"The global response to COVID-19 saw the most rapid and extensive vaccination rollout in history."*

(5) Rather than stating that vaccines were implemented across "all geographies," consider being more specific.

Response: As noted above, we have adjusted the first sentence and no longer use the phrase "all geographies".

(6) Are the findings genuinely novel? If not, highlight what's new about this analysis. Have no other studies on VE in low-income settings been conducted?

Response: Although limited by word count constraints in the Abstract, we now highlight the data linkages that form the key novelty of this analysis as follows: *"To systematically quantify these disparities, we generated novel linkages across large public datasets..."*

Main Text:

Global Disparities in Vaccine Introduction and Scale-Up:

(7) Provide a definition for GNI.

Response: We now define GNI per capita as follows: *“Among 204 countries with World Bank data on per capita Gross National Income (GNI),...”*

We elaborate further in the Methods as follows: *“Country-level income status was obtained from World Bank estimates of per capita Gross National Income (GNI) based the Atlas method?”*

(8) Does the study need an ethical statement clarifying that all study data is publicly available, aggregated by country, and non-identifiable?

Response: We address this under the Data Sources subheading as follows: *“All data used in this study are publicly available, aggregated by country, and contain no individual-level information.”*

(9) In Lines 33 to 39, the sentences seem contradictory: the first states that there’s robust protection and no significant differences in VE at the country level, while the next sentence mentions stark disparities across the globe.

Response: This point is well taken. There is a discrepancy here between the protection conferred by the vaccines (where we saw no significant disparities) and their implementation and evaluation (where we saw significant disparities). We have modified the phrasing to ensure the contrast is clear: *“Our findings highlight the strong protection conferred by COVID-19 vaccines across diverse populations. Nonetheless, our results quantify the stark disparities that pervaded each stage of the global COVID-19 vaccine rollout, and highlight key evidence gaps related to products and platforms being used across much of the globe.”*

(10) In Lines 41 to 45, the two sentences also appear contradictory. Instead of stating that vaccines were widely implemented in December 2020 and then highlighting global disparities, perhaps provide context. For example, mention that vaccines were distributed widely in high-income countries (HICs) while low- and middle-income countries (LMICs) lagged behind. Cite examples of later Covid-19 vaccine implementation in LMICs.

Alternatively, start with Line 45 and link assessments of VE with the vaccine rollout, noting the discrepancies between HICs and LMICs.

Response: To clarify the phasing and avoid potential contradiction, we have expanded this introductory text as follows: *“Following the onset of the COVID-19 pandemic, vaccines capable of reducing the risk of SARS-CoV-2 infection, disease, and severe outcomes were developed at unprecedented speed and introduced worldwide from December 2020^{1,2}. The ensuing vaccine rollout was characterised by marked global disparities in scale and timing, with low- and middle-income countries (LMICs) facing significant delays in vaccine access compared to high-income countries (HICs)^{3,4}.”*

(11) "Minimum vaccination date" doesn't make sense. Did you mean "earliest vaccination date"?

Response: We have changed “minimum” to “earliest” as suggested: *“We defined introduction as the earliest recorded vaccination date in WHO or OWID.”*

(12) The sentence "had recorded vaccinations" could be improved for readability. Perhaps rephrase it as "had recorded at least one vaccination." Additionally, the median time to vaccine introduction might be clearer if stated as: "The median time to vaccine introduction was 20 days (IQR 15-41) in countries with the highest GNI quartile, 57 days (21-77) in the second highest, and 78 (53-95) and 103 (91-124) in the second lowest and lowest quartiles, respectively."

Response: We have modified the phrasing as suggested to *“countries had recorded at least one vaccination”*.

We have shifted our analysis of time to introduction to focus on dates rather than differences relative to 8th December 2020. This avoids the necessity of anchoring analyses to a single approval date (as noted in comment 15).

As suggested, we have modified the phrasing of the sentence on vaccine introduction to improve readability: *"The median vaccine introduction date – reflecting the point at which 50% of countries had recorded at least one vaccination – was 30 December 2020 among countries in GNI quartile 4 (highest income), 26 January 2021 in quartile 2, 24 February 2021 in quartile 3, and 15 March 2021 in quartile 4 (lowest income; Fig. 2B; see Supplementary Table 1 for IQRs)."*

(13) In Line 45, a smoother transition between the first two sentences on vaccine implementation and effectiveness is needed. This could be achieved by linking the ideas earlier, so the transition isn't as abrupt.

Response: We have extended the Background section to introduce each aspect of the study in more depth. Sections on vaccine implementation and vaccine effectiveness are now covered in separate paragraphs to avoid an abrupt transition mid-paragraph.

(14) The mean time reporting in Line 65 is confusing with the different days and numbers in parentheses (20, 57, 78, 103). These need better labeling or clearer explanation.

Response: See response 12 – we have now updated the analysis to focus on dates.

(15) Justify the choice of approval date in Line 63. For example, while December 8th marks the first vaccine administered in the UK, note that vaccines were approved earlier.

Response: See response 12.

(16) What does "moderate consistency" mean? Clarify by specifying the agreement between WHO and OWID databases. For example, explain which data source was used for the median time estimates and how alignment with the other database affected the results.

Response: We have modified the wording to avoid the subjective statement "moderate consistency". We now focus on quantifying the agreement as follows: *"When comparing WHO and OWID databases, 186/203 (92%) countries had records of vaccine introduction in both databases, of which 107/203 (53%) dates were within ± 7 days (median [IQR] difference of 6 [2–16] days)."*

As noted in the Methods, we have used both data sources to establish introduction dates, *"taking the earliest date from the WHO and OWID databases."* Our aim here is to take advantage of the full complement of available information on vaccine introductions, and we have added the following wording to reflect this: *"thereby leveraging the complementary data flows of these two resources".*

(17) Define "primary series" clearly.

Response: We have added a definition of 'primary series' in the Methods as suggested: *"The definition of 'primary series' is specified by the product-specific use authorisation and can vary by product and country. In most cases, this represents two doses, although the Ad26.COV2.S vaccine (widely implemented as a single dose primary series) is a notable exception."*

We also included a brief definition of "primary series" in the Results as follows: *"The milestone of 40% primary series coverage by 31 December 2021 (with 'primary series' defined by the product-specific use authorisation)"*

(18) The milestone mentioned seems like an afterthought; provide more details about it.

Response: Thank you for this suggestion. We have brought the key supporting reference outlining these milestones in earlier in the paragraph as follows: *"We then explored the scale-up of vaccine implementation based on several coverage milestones specified in the 2021 WHO global COVID-19 vaccination strategy¹⁰."*

(19) Identify which low-income country performed well and whether there are lessons to be learned from it.

Response: Our primary aim is to systematically quantify global trends rather than focusing on individual countries. However, we agree that it is beneficial to highlight LMICs that were early to

achieve coverage milestones and to cite relevant commentaries discussing how this was achieved. We have therefore added the following to the discussion: *"In GNI quartiles 1 and 2, the earliest countries to reach the milestone of 40% population coverage were Mongolia, Bhutan, Cambodia, and Ecuador. As detailed further in country-specific commentaries²⁶⁻²⁹, swift scale-up in these countries was associated with early purchasing of vaccines outside of the COVAX facility (including from manufacturers based in China and India), strong existing immunisation infrastructure, and early investment to expand implementation capacity."*

(20) Clarify the meaning of "without date restriction" and ensure the numbers are presented relative to GNI groups.

Response: As suggested, we have rephrased the sentence to clarify the interpretation: *"Among countries that achieved 40% coverage (either before or after the milestone target of 31 December 2021), the median time from vaccine introduction to this coverage threshold was 190 (160–209), 239 (182–274), 272 (224–356), and 450 (347–637) days, respectively."* The order of GNI quartiles reflects that used in the preceding sentence.

(21) Lines 63 to 70 could be presented more clearly.

Response: See response 12, which specifically relates to these lines.

(22) Lines 80 to 84 have a similar issue. Reporting styles like "respectively" can disrupt readability, and the data might be better displayed in a table.

Response: Thank you for this comment. We appreciate that different audiences will find different formats intuitive. These data are presented visually in **Fig 2** and in a tabular form in **Supplementary Table 1**. Within the text, our view is that the 'respectively' formatting style is an appropriate and succinct way of presenting the quartile-specific estimates in sequence, though we defer to the editor for guidance on preferred formatting.

(23) In Line 92, the phrase "In the context of limited supply" seems circular. Rephrase to clarify the impact on high-risk groups.

Response: We have simplified the sentence to avoid circularity: *"In the context of limited supply, vaccinating high-risk groups including older adults and healthcare workers was recommended as part of the WHO roadmap for COVID-19 vaccine prioritisation⁷."*

(24) In Lines 92 to 100, explain whether the call to vaccinate high-risk groups (e.g., the elderly and healthcare workers) was consistent across countries. The discrepancy between 10% vaccination coverage for older persons and 56% for healthcare workers in the lowest quartile needs more context.

Response: Although implementation policy varied by country, the recommendation to prioritise older adults and healthcare workers was laid in the WHO prioritisation roadmap – first published in October 2020. We have adjusted the text to clarify this (see updated wording in response 23 above).

To provide more context on interpretation of these data, we have added details on reporting completeness by GNI quartile alongside the proportion of the population within high-risk categories:

- For healthcare workers: *"Notably, data on coverage in healthcare workers were available for 122/203 (60%) of countries (see **Supplementary Table 2** for breakdown by GNI quartile), and among those reporting, the population accounted for by healthcare workers varied from 5.4% (3.4–6.9%) in GNI quartile 4 (highest income) to 0.4% (0.3–0.8%) in GNI quartile 1 (lowest income)."*
- For older adults: *"Data on coverage in older adults as of December 2021 were available for 127/203 (63%) countries. Where defined, the threshold used to define older adults varied between 50 and 75 years, with a skew towards younger thresholds in lower-income quartiles (**Supplementary Table 3**). Among countries reporting, the proportion of the total population accounted for by older adults varied from 25% (17–27%) in GNI quartile 4 (highest income) to 9% (8–11%) in GNI quartile 1 (lowest income; **Supplementary Table 2**)."*

Full data are reported in **Supplementary Table 2**.

Notably, the changes above highlight that the target denominators for healthcare workers in the lowest-income quartile (median 0.4%) are substantially smaller than that for older adults (median 9%), thus providing the additional context requested by the reviewer.

(25) Is the median of 100% coverage for healthcare workers in the second-lowest income group correct? Could there be biases in the data?

Response: The calculated median is 'correct' in the sense that it is the median value among the reported coverage estimates. However, the reviewer is right that this is likely to reflect a biased estimate of real-world coverage, and we certainly agree that the results should be interpreted with caution. We now draw attention to this issue as follows:

- We clarify the gaps in the available data in the Methods as follows: "*Healthcare worker definitions may vary by country and were not included in the database.*"
- We discuss the potential data qualities (and specifically reference the 100% coverage in quartile 2 mentioned above) in the Discussion as follows: "*Criteria used to define healthcare workers were not available and age thresholds to define older adults were frequently missing. The high coverage in these populations (e.g., 100% median coverage among healthcare workers in GNI quartile 2 as of December 2023) may reflect an underestimation of the target denominator (defined by International Labor Organization statistics) relative to the coverage numerators captured by WHO reporting systems.*"

(26) How are "older adults" defined in each country? Include this in the supplement if necessary.

Response: We thank the reviewer for raising this point. As suggested, we have added **Supplementary Table 3** tabulating the definitions used. We cite this in the text as follows: "*Where defined, the threshold used to define older adults varied between 50 and 75 years, with a skew towards younger thresholds in lower-income quartiles (Supplementary Table 3).*"

(27) Clarify what specific differences were smaller, as it's not clear which data comparisons are being made.

Response: We made the comparison explicit as follows: "*Disparities in vaccine coverage among healthcare workers and older adults across GNI quartiles were smaller than the disparities observed in population-wide coverage*"

Global Differences in the Evaluation of Vaccine Effectiveness (VE):

(28) Provide a clear definition of VE used in the study.

Response: We now define VE as follows: "*We obtained studies of absolute primary series VE – which measures the reduction in SARS-CoV-2 infection or disease among vaccinated compared to unvaccinated individuals – from published...*"

(29) In Line 121, when introducing the section on global differences in VE evaluation, explain what VIEW-hub is and how you selected the studies.

Response: As requested, we provide a brief introduction to VIEW-hub in the main text as follows: "*...the VIEW-hub living literature review conducted by the Johns Hopkins School of Public Health International Vaccine Access Center (IVAC)⁹. The VIEW-hub database collates VE studies meeting pre-specified criteria of methodological rigour, including adequate accounting for confounding variables and inclusion of laboratory-confirmed outcomes.*"

Additional details are provided under Data Sources in the Methods section.

(30) Line 127 is missing "respectively" or a similar word for clarity.

Response: We have reviewed the text, and the word "respectively" is included as follows: *"Of these, 345 (82%), 64 (15%), 9 (2%), and 5 (1%) were from GNI quartiles 4 (highest income), 3, 2, and 1 (lowest income), respectively."*

(31) Has a comprehensive review of VE studies been conducted before? Clarify if this is new.

Response: We now provide a clearer introduction to the scope of this analysis as follows: *"We next sought to systematically quantify disparities in the availability of VE data for different products and platforms."*

We also clarify the key novelty of our approach in the Discussion as follows: *"Previous studies have highlighted an unequal distribution of COVID-19 research²²⁻²⁴. However, the linkage of a comprehensive public database of VE estimates to country-level income status is a key novelty of our study, and enabled us to quantify inequities in the timing and platform distribution of VE data availability."*

(32) In Line 133, change "quartiles" to "GNI quartiles."

Response: We have modified the sentence as suggested: *"VE data for GNI quartiles 3, 2, and 1..."*

(33) When you mention that a country had a VE study, clarify whether this means the study examined VE in the country's population or whether the authors were from that country.

Response: We have amended the phrasing to "VE estimate" rather than "VE study" to better clarify that it is the location of the study that we are concerned with here: *"Among countries with at least one published VE estimate..."*

Please also note that this point is emphasised during our introduction of the VE database: *"As of 11 January 2024, the database contained 425 studies with country-specific VE estimates..."*

(34) In the paragraph starting in Line 148, you seem to imply there are multiple VE metrics. Clarify the differences between these metrics.

Response: We now clarify that we are distinguishing between metrics corresponding to the severity of disease endpoints as follows: *"Studies reported VE estimates for outcomes of varying severity (with individual studies often reporting separate estimates for multiple outcomes)."*

(35) Provide more details on why VE might differ due to demographics, comorbidities, or social mixing for each VE measure. Non-pharmaceutical interventions (NPIs) should also be discussed.

Response: As described in our response to comment 3, we now provide a more detailed rationale for exploring geographic variation in VE in the Background section. We reiterate this rationale and highlight the contribution of NPIs as follows: *"...due to differences in demographic and clinical factors that impact the strength and duration of vaccine responses⁵⁻⁸, comorbidity rates, vaccine eligibility criteria, social mixing patterns, and non-pharmaceutical interventions."*

(36) Lines 163 to 165 seem like a major point. Consider moving it to the discussion and expanding on it there.

Response: These lines provide the key rationale for our meta-regression, and we therefore favour their inclusion at this point in the manuscript. However, we currently pick up this key point at several points in the Discussion section, including:

- *"Together, these findings suggest that the protection offered by COVID-19 vaccines against severe disease is generalisable across populations with diverse demographic and clinical characteristics."*
- *"The gaps in VE data have presented a significant challenge to policy-makers – key decisions relating to the use of COVID-19 vaccines in LMICs have required the extrapolation of findings from mRNA vaccines being administered in high-income countries."*

(37) In Line 171, "meta-regression" is introduced too late. Prime the reader earlier in the report.

Response: Thanks for this suggestion. We now prime readers in the Abstract as follows: *“For vaccines with available VE data across multiple income settings (i.e., BNT162b2, mRNA-1273, and ChAdOx1-S), our meta-regression highlighted...”*

We also bring in the concept of meta-regression earlier in this section of the results: *“The availability of data from different GNI quartiles provides an opportunity to explore potential geographic differences in VE based on the emerging global evidence base. To this end, we performed a meta-regression to assess potential variation in VE by country income status.”*

(38) How were populations within studies accounted for? If VE varies due to characteristics like NPIs and contact patterns, wouldn't VE estimates also vary widely within populations?

Response: We included VE estimates derived from the general population, subject to the constraints described in comment 3 and the inclusion criteria (e.g. minimum confounding adjustment) applied by VIEW-hub). We also excluded VE estimates derived from specific subpopulations (e.g. healthcare workers, immunocompromised populations) that are unlikely to be representative of the general population

Nonetheless, we acknowledge that our meta-regression is tailored to capture broad geographic trends in VE, and that additional factors may shape VE within populations. Our multi-level modelling approach highlighted the high variability among VE estimates within countries, and we draw attention to this at relevant points in the results (e.g. *“79% of variation across VE estimates was attributed to within-country heterogeneity”*).

To draw further attention to this issue, we have modified the Discussion to highlight the need for further exploration of within-country variability as follows: *“However, variability in variant-specific VE estimates was greater within countries than across countries. This likely reflects differences in populations studied, follow-up duration, and outcome definitions, among others. These findings highlight the need for tailored analyses, such as stratified VE estimates within key subpopulations, to better understand the factors shaping VE within countries.”*

(39) Explain how pooled VE estimates were calculated, particularly considering different study population sizes.

Response: Thank you for this comment. The Methods section provides a detailed explanation of the three-level modelling approach. We have extended this section to clarify how pooled VE estimates were calculated: *“Pooled risk ratios for each GNI quartile were calculated based on the three-level models, then converted to VE estimates by taking $[(1 - \text{pooled risk ratio}) * 100]$.”*

The precision of VE estimates is driven primarily by the number of outcome events included rather than the size of the study population (though the two are typically related). We have added details in the Methods to clarify the weighting of study estimates within the meta-regression models: *“with weights assigned to each estimate based on their standard errors.”* This weighting ensures that larger studies with lower variability contribute more to the pooled estimates.

(40) Given that VE wanes over time, how might the timing of studies impact VE calculations? Add this as a limitation if necessary.

Response: We were conscious of the potential impact of waning VE during the design of our analysis, and accounted for this in several ways:

- Where studies reported on a series of post-vaccination periods (e.g. days 1-180, days 180-360...), we excluded later periods that would be more subject to waning. This is clarified in the Methods as follows: *“If studies reported separate VE estimates only for sequential sub-periods, we excluded later periods to mitigate the effect of waning.”*
- We report on variation in study duration in **Supplementary Table 6** and in the text as follows: *“The median (IQR) for the upper limit of follow-up (the maximum potential duration of follow-up among study participants) after primary series vaccination was 25 (21–35) and 28 (16–48)*

weeks for GNI quartiles 4 and 3, respectively, and 38 weeks for the single estimate from GNI quartile 2."

- We conducted a sensitivity analysis in which we restricted our meta-regression to studies with a maximum 6-month duration: *"Our findings were consistent in a sensitivity analysis limited to VE estimates with a maximum upper limit of follow-up of 26 weeks (Supplementary Table 7)."*
- We include a specific limitation highlighting the potential impact of waning: *"Finally, due to the impact of waning immunity on VE, we report on variation in the maximum potential follow-up time across studies, although follow-up may have been shorter for many individuals enrolled in each study. However, our findings were consistent in a sensitivity analysis restricted to the first 26 weeks after primary vaccination."*
- Finally, the stratification of results by variant context also implicitly accounts for the timing of studies.

We feel that the measures above constitute a robust approach towards accounting for waning VE within the constraints of the publicly available datasets.

(41) In Line 180, clarify what is meant by "upper-limit of follow-up."

Response: We have clarified the definition as follows: *"The median (IQR) for the upper limit of follow-up (the maximum potential duration of follow-up among study participants)..."*

(42) Line 182 reports no significant differences in VE but also mentions variations. Can these variations be explored in greater depth?

Response: See response 38, which addresses this point. Specifically, a detailed exploration of within-country heterogeneity is beyond the scope of our study, but we highlight the need for tailored analyses in the Discussion as follows: *"These findings highlight the need for tailored analyses, such as stratified VE estimates within key subpopulations, to better understand the factors shaping VE within countries."*

(43) In Line 184, what is I²? Provide an explanation.

Response: We now provide an explanation of the I² statistic as follows: *"Of the observed variation in VE estimates not attributable to sampling error (quantified via the I² statistic), 55% was attributed to differences between countries (I² level 3), while within-country heterogeneity accounted for 43% (I² level 2)."*

We also provide further details on I² in the Methods section as follows: *"Additionally, we used the I² heterogeneity statistics to quantify the proportion of the variation in effect sizes attributable to heterogeneity within countries (I² level 2) and between countries (I² level 3)¹⁰."*

(44) The definition of VE is confusing in the context of both severe Covid-19 and Covid-19 mortality. Wouldn't these require two different VE metrics? How are they combined?

Response: See response 34, in which we have clarified that VE estimates against outcomes of varying severity were considered separately. We have amended the Methods to make this explicit: *"We stratified analyses by product, outcome severity, and dominant circulating variant."*

In practice, no comparisons of VE against COVID-19-related mortality met our eligibility criteria for meta-regression, as we clarify in the Results section: *"...our outcomes of interest were severe COVID-19 and COVID-19-related mortality, although no comparisons of the latter were eligible for inclusion (Supplementary Table 4)."* As such, our final analyses are limited to VE against severe disease.

(45) Remove the summary in Lines 206 to 208 and begin with a discussion instead.

Response: We have shortened this sentence and amended it to highlight the novelty of our study. The final wording is as follows: *"We used novel data linkages to systematically quantify global disparities in the implementation, scale-up, and evaluation of COVID-19 vaccines, including disparities in time to first*

introduction, vaccine platform use, interim coverage targets, and real world evaluation of vaccine performance."

(46) In Line 283, add a period between "research" and "The VIEW-hub."

Response: Thank you – we have amended as suggested.

(47) Lines 282 to 287 should introduce the "Global differences in..." section more clearly.

Response: Our intention in this section is to discuss the strengths of our study, and we have introduced and extensively discussed the global disparities in implementation and evaluation in the preceding paragraphs. We now make this shift more explicit as follows: "Strengths of our study include the novel linkages made across large public datasets and..."

We have also slightly restructured the Discussion, bringing the paragraph on implementation disparities (including the summary of COVAX) up to ensure that the key findings are covered in a logical manner that reflects that of the Results section.

(48) Remove the introductory sentence in Lines 307 to 310 and start with a statement on vaccine rollout disparities and the implications of inequities in implementation.

Response: Thank you for this suggestion. We have modified the start of the Conclusion to draw more attention to the implications of vaccine rollout disparities: "Notwithstanding the speed and breadth of the global rollout of COVID-19 vaccination, the rush to secure vaccines and protect populations led to stark inequities in implementation. In LMICs, earlier availability of vaccines and higher uptake could have averted a significant proportion of COVID-19 morbidity and mortality, along with associated economic and health system impact."

(49) Line 495: Include a disclaimer regarding any exclusion criteria used in the VE studies and provide more detail on this.

Response: We assume the reviewer is referring to inclusion/exclusion criteria applied in our handling of VE estimates rather than the specific inclusion criteria for participants of the individual studies (which would vary on a case-by-case basis relating to the specific objectives, population, and design of each study).

As suggested, we have added a disclaimer regarding the meta-regression eligibility criteria in the Methods section: "Among the VE estimates included in our descriptive analysis, we applied further inclusion and exclusion criteria based on VIEW-hub metadata to ensure a minimum degree of comparability required for a meta-regression." The specific criteria (e.g. limiting to homologous series, stratification by variant, etc) are provided in detail in the sentences that follow.

(50) Grouping GNI quartiles might result in arbitrary classifications. Discuss the limitations of using this method.

Response: Thank you for raising this point. To complement our analysis of GNI per capita quartiles, we have now added correlation plots that handle GNI per capita as a continuous variable (see reviewer 3 response 1 for further details). The most intuitive approach is likely to vary for different audiences, so we hope that including both will enhance the overall accessibility of our work.

(51) A table listing all VE estimates by study would improve reproducibility and indicate which estimates were used or excluded.

Response: All data and underlying code are available via our Github repository, but to improve accessibility we have added a supplementary dataset that includes VE estimates and relevant metadata. We cite this in the paper as follows: "A total of 51 studies met the eligibility criteria for inclusion in our meta-regression (Supplementary Data 2)."

(52) Could a country's vaccine dosing schedule (e.g., two doses three months apart versus two doses six months apart) impact VE? Discuss this.

Response: See response 1 above in which we directly address this point.

Discussion:

(53) Strengthen the discussion by expanding on the significance of findings, such as the disparity in VE estimates across income groups. Address how these disparities impact future vaccine rollouts and effectiveness evaluations.

Response: We have expanded on the suggested points throughout the Discussion.

Key changes are as follows:

- On geographic disparities in VE: *"Our findings are consistent with phase 3 trial data for BNT162b2, in which short-term efficacy estimates against symptomatic COVID-19 were similar in the USA, Argentina, and Brazil³². Early efficacy data for ChAdOx1-S were also similar in the UK and Brazil³³. However, variability in variant-specific VE estimates was greater within countries than across countries. This likely reflects differences in populations studied, follow-up duration, and outcome definitions, among others. These findings highlight the need for tailored analyses, such as stratified VE estimates within key subpopulations, to better understand the factors shaping VE within countries."*
- On future effectiveness evaluations: *"Lessons learned from trying to set up those VE studies, such as the efficiency and speed of leveraging existing respiratory disease surveillance platforms, should be applied to future pandemic preparedness and vaccine introduction planning to ensure that policy decisions are informed by VE estimates from diverse settings."*
- On future rollouts: *"The inequities in vaccine implementation and VE research observed for COVID-19 serve as a benchmark against which to measure progress in future pandemics. In the fall-out of COVID-19^{24,58}, we must lay the foundations to do better next time."*

(54) In Lines 213 and 216, discuss the role of mRNA vaccine costs in wealthier countries and how this may have contributed to disparities. Expand on the variation in coverage disparity and its implications.

Response: We have now expanded the second paragraph of the Discussion to cover the range of factors contributing to coverage disparities as well as their public health implications: *"The inequities in vaccine introduction and scale-up left LMICs vulnerable to higher rates of COVID-19 transmission and mortality, and all countries at increased risk of new waves caused by emerging strains²¹. The political, economic, and logistical factors underlying these inequities are highly multifaceted, with each country charting a different course through the process of negotiation and implementation. Key factors contributing to the broad trends quantified here include: the concentration of vaccine manufacturing capacity in high-income countries; monopolisation of vaccine supplies via bilateral purchasing agreements between high-income countries and vaccine manufacturers^{22,23}; and the recognition by late 2021 that booster doses were necessary to reinforce protection in the context of waning immunity and emerging variants, leading high-income countries to retain available doses for booster campaigns rather than distribute them to LMICs for primary vaccination²⁴."*

(55) In Line 241, relate comments about VE in HICs to studies and consider adding a citation.

Response: As suggested, we now provide indicative examples of countries and corresponding citations at the end of this sentence: *"...and VE study platforms and established methodological expertise for other pathogens like influenza^{39,40} (e.g., Israel⁴¹, UK¹, USA⁴², and Denmark⁴³)."'*

(56) In Line 291, could sensitivity analyses be performed in data-rich settings like GNI3/4?

Response: In practice, it is only GNI per capita quartile 4 that could be considered data-rich. As summarised in **Supplementary Table 3**, Q2 and Q3 make up a minority of studies for almost all eligible analysis (e.g. for the analysis of BNT162b2 VE against the Omicron variant, 1 study was from Q2, 5 were from Q3, and 18 were from Q4).

(57) The concluding section should address the "what's next" question. How does this research contribute to future studies or policy actions?

Response: We agree that the conclusion should be more forward-looking. We have edited the section to reduce repetition and highlight that vaccine inequities observed for COVID-19 (as documented in our study) should serve as a benchmark for assessing future progress. The updated wording is as follows:

"Notwithstanding the speed and breadth of the global rollout of COVID-19 vaccination, the rush to secure vaccines and protect populations led to stark inequities in implementation. In LMICs, earlier availability of vaccines and higher uptake could have averted a significant proportion of COVID-19 morbidity and mortality, along with associated economic and health system impacts. Disparities in VE research have created key evidence gaps for policy-makers and hindered context-specific optimisation of vaccine programmes. The inequities in vaccine implementation and VE research observed for COVID-19 serve as a benchmark against which to measure progress in future pandemics. In the fall-out of COVID-19, we must lay the foundations to do better next time^{24,58}."

Methods:

(58) Highlight how the study's methodological decisions align with Cochrane guidelines (Line 491).

Response: We included this sentence as our use of inclusion/exclusion criteria align broadly with Cochrane guidelines for meta-regression, and the relevance of multilevel models for handling hierarchical data aligns with Cochrane handbook section 16.8.2. However, we acknowledge that the reference to Cochrane guidelines is potentially confusing outside of the context of a full systematic review and have therefore deleted this sentence.

(59) In the file "WeeklySummary_COVID19_VE_Studies_20240111.xlsx" in the "Primary Series Studies" tab, the variables "dose_number" and "dose" are unclear. How do they fit into the definition of the primary series? If VE estimates are mixed between one or two doses, how will this impact the study's results?

Response: Please note the 'Description of Contents' tab that includes definitions of each variable, as provided by VIEW-hub. Specifically:

- dose number: "Denotes the number of doses received by the study participant (1 or 2 for two-dose vaccine regimens and 1+ for single dose regimen)."
- dose: "Denotes if the dose evaluated is the final ("final") dose in the primary series being evaluated (dose 1 for 1 dose vaccines, dose 2 for 2 dose vaccines, etc.); if the dose being evaluated is not the final dose, the dose number is provided."

As notes in response 51, we have added a supplementary dataset (**Supplementary Data 2**) that includes VE estimates and relevant metadata. This includes a 'Dictionary' tab that includes the above definition for 'dose_number'.

As noted in response 17, we now provide a definition of 'primary series' in the Methods and Results sections.

As noted in response 1, we included measures to ensure comparability among estimates included in meta-regression, including standardisation of the number of doses among VE estimates included in a given analysis.

Figures:

(60) Figure 1: Consider reorganizing the figure for better clarity and resolution. Using a full-page layout or separating the subfigures could improve readability. Specifically, breaking the maps (A) into their own figure and grouping other components separately would help. The current resolution is low, and small world maps are hard to interpret.

Response: We thank the reviewer for this suggestion. As suggested, we have separated out the maps into their own figure (**Figure 1**) and present the remaining charts separately (**Figure 2**). We have also

increased the resolution of the figures. We agree that this has added clarity to the presentation of our findings.

(61) In 1A, the grey shading for "not reached" is difficult to distinguish. Darker greys (e.g., for Greenland and Syria) are more visible.

Response: Thank you for pointing this out. We now use darker grey shading as suggested.

(62) In 1B, with ~195 countries divided into quintiles, can the countries in each quintile be listed somewhere (e.g., in a supplement)?

Response: In **Supplementary Fig. 8**, we have included a map of countries shaded by GNI per capita quartile. We have also added a supplementary dataset (**Supplementary Data 1**) that includes key country-level metrics underpinning this manuscript (see reviewer 3 response 1). GNI per capita and associated quartiles are included in this dataset.

(63) Figure 2: Clarify whether you mean the "whole population" or the target group within each country, which can vary. If a country expanded vaccination to children under 18, the coverage rate could be skewed. Make sure the numerators and denominators are meaningful for cross-country comparisons.

Response: Denominators represent the whole populations and are therefore not subject country-specific vaccine policies. We now clarify this in the figure legend as follows: *"Whole population denominators reflect estimates of overall country population size by the United Nations Population Division."*

(64) Figure 3: This figure also needs to be broken down. Presenting subfigure (A) separately and grouping the others logically would enhance clarity.

Response: We thank you for your valuable suggestion to improve the clarity of this figure. We have reorganised the figure by presenting the map - subfigure (A) - as part of **Figure 1** and the remaining panels as part of **Figure 2**.

(65) Figure 4: Increase clarity by enlarging the figure and possibly organizing it in a grid. Remind readers why no data is shown for GNI quartile 1. Could the lower VE values in GNI quartile 2 be due to study timing during the Omicron wave, which lowered VE?

Response: Thank you for these suggestions. As noted in comment 1, Ad26.COVID2.S was dropped from the final analysis after an error in the analysis pipeline was rectified. In light of this, we have enlarged the three remaining plots and thereby improved clarity.

As noted in reviewer 3 response 17, we have updated our analyses to use World Bank GNI data defined by the Atlas metric, enabling data for several additional countries to be included. As a result of this change, GNI quartile 1 is now represented in the figure, albeit only for the analysis of ChadOx1-S in relation to the Delta variant. We clarify the absence of data for GNI quartile 1 for the remaining analyses in the figure legend as follows: *"The absence of data for GNI quartile 1 for all but one comparison reflects the absence of estimates for this quartile that met the meta-regression inclusion criteria."*

Although the timing of the Omicron wave was not synchronised across countries, our meta-regression focused on VE estimates relating to the initial period of protection after vaccination (as discussed extensively in response 40).

Reviewer #1 (Remarks on code availability):

(66) The code was scanned over and seemed appropriate. However, the data used for the analyses seems to compare countries "like-for-like" when in fact differences in the definition of primary series seems to occur (i.e. 1 v 2 doses).

Response: We thank the reviewer for taking the time to consider our analytic code. As noted in response 1, one VE estimate relating to 2 doses of Ad26.COVID2.S was erroneously included in our

previous analysis. This has now been rectified, thereby ensuring that all estimates included in a given comparison meet a minimum degree of comparability (specifically, consistency of product, number of doses, outcome severity, and variant). See comment 1 for further details.

Reviewer #2 (Remarks to the Author):

Response: We support this initiative and are grateful for your constructive feedback.

Reviewer #3 (Remarks to the Author):

Summary

Our read of the work is it intends to highlight the correlation between national economic performance and vaccine program performance. The work concludes that higher gross national income per capita (GNI) correlates with better vaccine emergency response program measures: earlier initial doses, earlier achievement of coverage milestones, and earlier post-distribution evaluation of the response technology. This work also includes a meta-analysis of vaccine efficacy (VE) to assess if that varies with GNI, and concludes that it does not. Were such claims clearly proven and well-quantified, that would be useful information for future public health planning efforts, and thus significant.

However, we are concerned with overall methodology and especially the analytical approach to the data. We think these concerns undermine the potential usefulness of this work, but that those concerns can potentially be remedied. As such, we have several suggested improvements. Briefly, we think this work would be improved by:

- Presenting correlation plots and calculations rather than quartile visualizations. Linear and rank regression plots would be clearer about the overall quality of correlation between GNI and the various vaccine metrics.
- Dropping the meta-analysis of VE. This muddles the apparent research question, which seems to focus on program performance. We agree with assessing the timing and completion of evaluation work as a useful indicator of the overall emergency response program, but the VE comparison seems to be distinct work and likely requires different statistical machinery to do properly.
- Clarifying the status of evidence for various conclusions, both as found in this and other work, and in particular what measures had a demonstrated causal chain versus observed correlations.

Lastly, we are reading this work as researchers that focus on infectious disease modeling. We could imagine using this kind of work as, say, the basis for selecting archetypal settings in a scenario analysis. The calculations and visualizations presented in this paper could be useful for understanding the observed variation in the data, but of course would not necessarily match those hypothetical analytic needs (e.g. we might be interested in 3 groups instead of 4). For our kind of use case, the most valuable result would be a well curated, documented, and organized data set; that seems to be roughly present, but in need of refinement and highlighting.

(1) Below, we expand on the summary points above, and conclude with some detailed notes.

Research Questions & Findings

Our interpretation of the work is that it intends to evaluate the correlation between economic performance (as indicated by GNI) and vaccine programme performance (as indicated by timing of initiation, of achieving key coverage milestones, and of evaluating programme benefits). The work also intends to investigate correlation between GNI and assessed VE.

The work concludes that GNI and various programme performance indicators are correlated. This is argued by visualizing the difference in vaccine programme distributions between GNI quartiles. The work also concludes that GNI and VE are not correlated, by illustrating that the pooled VE estimates have overlapping uncertainty intervals, and checking that covariates are insignificant.

How did were those questions asked and results found?

The analysis combined a variety of country-level data sources on income and vaccine program implementation. GNI per capita for 195 countries was obtained from the World Bank. The date of vaccine introduction was taken as the earliest date listed in the WHO and OWID databases and was available for 194 countries. Data on primary series vaccine coverage was obtained from the OWID database for all but three countries/territories.

The analysis compares 1) timing of vaccine introduction, 2) scale up of vaccination, and 3) evaluation of VE. The metric used to assess introduction timing was minimum recorded vaccination date. The metrics used to

assess scale-up were 1) whether a country achieved 40% coverage by December 31, 2021, 2) time to achieve 40% coverage, and 3) coverage (stratified by risk group) as of December 2021 and 2023.

The metrics used to assess VE evaluation were 1) number of studies available, 2) time from introduction until first study publication, and 3) percentage of studies stratified by vaccine platform. To perform this evaluation, the analysis considers published, preprint and gray literature sources from the VIEW-hub literature review from Johns Hopkins IVAC. Studies with multi-country VE results and/or those from countries which lacked available GNI data were eliminated, resulting in 422 (out of 434) studies used for the authors' descriptive analysis. For example, studies from Cuba and Taiwan were eliminated because World Bank GNI data was not available for these countries. Overall, VE data was available for 47 out of the 192 countries reporting vaccine introduction.

Differences across countries were observed by reporting these metrics by GNI quartile and by plotting boxplots of each metric stratified by GNI quartile. Additionally choropleths were utilized to show variations in these metrics by country. No statistical tests were performed to quantify the significance of these differences.

A meta-regression was performed with the intent to assess differences in VE by GNI. Analysis was stratified by vaccine product and variant. 53 out of the 422 studies used for the descriptive analysis were selected for this meta-regression based on criteria such as virus variant, outcome of infection, and number of studies by GNI quartile.

Should we believe it?

The current analysis relies on looking at differences across income quartiles to demonstrate the supposed association in vaccine program performance. However, the categorization into quartiles is somewhat arbitrary and does not fully use the data. Presentation of correlation plots and calculations in combination with regression calculations - that is, computing linear and/or rank correlation coefficients without stratifying into quartiles - would provide a more convincing argument and better use all the available data.

The analysis of vaccine efficacy seems distinct from the other research questions being addressed in this paper. The relevant question seems to be "are these GNI quartiles different?", but the meta-regression analysis does not seem suited to answer it. While not framed this way in the manuscript, we read the conclusion here as "we cannot clearly see if there is a difference". The analysis concludes that GNI as a meta-regression covariate is insignificant, but the more appropriate statistical comparison here would be "is the difference between strata clearly less than X%?". We expect the likely answer to that would be "no" for large enough X so as to be uninteresting, given how wide the measurement uncertainty intervals are here. However: we do not think this part of the analysis serves what we read as the overall aim of the work. We think the aim here is to understand relative operational pace in vaccine programs, which of course includes post-distribution elements like ongoing evaluation; the actual VEs concluded by those ongoing evaluation programs is a separate issue. A more relevant issue for VE studies would be something like whether studies meet minimal quality standards (which publication is not necessarily an indicator of).

Lastly, much of the language in the discussion and conclusion sections implies causal relationships. That language is generally not supported by the specific analyses here, nor by citations, and generally falls into the "everybody knows X" category. This is a framing error, which we find undermines the actual analysis and results that could be presented here.

Response: We thank the reviewers for their thoughtful critique of our paper. We acknowledge the validity of several of the concerns raised, although there are aspects where our perspectives differ.

We respond to the key critiques raised in this summary below.

On the value of a curated dataset

We thank the reviewer for this excellent suggestion. See comment 18 below for details on the new curated datasets provided with this revised submission. We hope that others will find value in these, as indicated by the reviewer.

On handling of GNI as a continuous rather than categorical data

Thank you for this suggestion. We recognise the value of exploring GNI per capita as a continuous variable and have added this as a secondary analysis. Plots showing the correlation between GNI and vaccine implementation and evaluation metric are now presented in **Fig. 3**, and Spearman's rank correlation coefficients are provided in the figure legend and main text. We highlight these findings in the text as follows:

Methods:

- *"As a secondary analysis, we used Spearman's rank correlation coefficient to assess the association between GNI as a continuous variable and: vaccine introduction; vaccine scale-up; and first VE publication."*

Results:

- *"A strong correlation between country income status and vaccine introduction data was also apparent when handling GNI as a continuous variable (Spearman's rho -0.66, $p < 0.0001$; 203 estimates; Fig. 3A)."*
- *"Disparities in vaccine scale-up were also apparent when considering GNI as a continuous variable (Spearman's rho -0.73, $p < 0.0001$; 146 estimates; Fig. 3B)."*
- *"Consistent with these disparities, there was a strong correlation between GNI as a continuous variable and the date of first VE research publication (Spearman's rho -0.42, $p = 0.003$; 48 estimates; Fig. 3C)."*

The most intuitive format for these data will vary for different audiences. We find the reported differences by GNI per capita quartile, as presented in **Fig. 2** and elsewhere in the main text, to be a powerful way of summarising disparities in vaccine implementation and evaluation (e.g. supporting statements such as the following: *"The time difference between the initial publication of VE data in the highest and lowest income GNI quartiles was over 9 months."*).

However, we acknowledge that others may find the continuous correlation plots more intuitive. Our aim is to make our study useful to a wide variety of audiences, and hope the inclusion of both approaches supports this.

Please also note our response to comment 4 below, wherein we have provided a curated dataset containing both continuous and categorical GNI per capita measures to support further analysis by others.

On inclusion and framing of the VE analysis

Our aim for this study was to explore whether there is any evidence of broad geographic differences in VE within the emerging global evidence base.

We acknowledge that the scope and motivation for this analysis needs to be expressed more clearly, and have revised the paper accordingly:

- We establish the biological rationale and public health importance of exploring variation in VE in the introduction: *"Protection may also vary according to demographic and clinical characteristics of the vaccinated population, and reduced immune responses and efficacy have been reported in LMICs relative to HICs for multiple vaccines⁵⁻⁸. This may limit the generalisability of VE findings across geographically disparate settings."*
- We establish the scope of the analysis in the relevant subheading of the Results: *"The availability of data from different GNI quartiles provides an opportunity to explore potential geographic differences in VE based on the emerging global evidence base."*
- We elaborate on issues associated with statistical power in the discussion section: *"Our meta-regression was underpowered to detect geographic differences in VE due to the limited data available for lower-income quartiles..."*

The reviewer is correct that different statistical machinery is required for these analyses, and we tailored our approach accordingly – applying a multi-level metaregression framework for the VE analyses as opposed to the descriptive metrics used in prior analyses. The statistical approach for our meta-regression aligns with previous broad-scale analyses aiming to synthesise COVID-19 VE data (e.g. Wu et al 2023; *Lancet Resp Med* 11, 439–452).

Similar approaches have also been used to explore variation in VE. For example, in the case of rotavirus vaccine, significant variation by country income status is evident based on a comparable study count to the present study (e.g. Sun et al 2021; *JAMA Pediatr* 175(7):e210347 and Burnett 2020; *Lancet Glob Health* 8(9):e1195-e1202), and the reporting of VE estimates by income status (as we do here) provides a valuable way to summarise this variation. In essence, had a similar gradient in VE been evident for COVID-19, our analytic approach would have been well suited to capturing it.

On causal relationships

Our intention was not to draw a causal line between lower income status and variation in implementation or VE. Rather, we aimed to quantify disparities using a clear and consistent framework. Our stated aims reflect this: *“In this study, we harnessed novel linkages across large public datasets to systematically quantify global disparities...”* and *“...we considered the association between country-level income status and...”*).

Having quantified significant disparities in vaccine implementation, we view it as reasonable and necessary to discuss key factors contributing to these trends, and we attempt to do this in the discussion section. Nonetheless, the reviewer’s point is well taken, and we acknowledge the importance of framing the discussion appropriately. We have amended our phrasing to introduce ‘contributing factors’ as opposed to causal explanations in the discussion: *“The political, economic, and logistical factors underlying these inequities are highly multifaceted, with each country charting a different course through the process of negotiation and implementation. Key factors contributing to the broad trends quantified here include: the concentration of vaccine manufacturing capacity in high-income countries; monopolisation of vaccine supplies via bilateral purchasing agreements between high-income countries and vaccine manufacturers^{22,23}; and the recognition by late 2021 that booster doses were necessary to reinforce protection in the context of waning immunity and emerging variants, leading high-income countries to retain available doses for booster campaigns rather than distribute them to LMICs for primary vaccination²⁴.”*

Later in the discussion, we have deleted a sentence referring to “The causes of this inequity...” in light of the reviewer’s cautionary note.

As suggested, we have also added citations throughout the discussion, both to broader policy-level analyses and individual country case studies. See response 6 for further details.

Overall, we thank the reviewer for these valuable critiques. Responding to them has strengthened the manuscript, and added depth and nuance to our discussion.

Detailed Notes

(2) Typos/grammar

Line 143: Missing closing parenthesis

Line 25: Stray “)”

Line 283: Missing period after “VE research”.

Line 448: Extended Data Fig. 12 is referenced, but only 8 extended data figures are presented.

Line 452: “Primary series coverage estimates of high-risk populations...”

Lines 518 - 526: Font size is smaller than the rest of the text.

Response: Thank you. We have fixed these typos in the revised manuscript.

Figures

(3) In Figure 1A it is difficult to distinguish between “not reached” and “90+”.

Response: Thank you for this comment. We changed the colours to make the distinction clearer.

(4) In Figure 2, shifts in coverage from December 2021 to December 2023 are impossible to distinguish given the layout - perhaps it would be better to show side-by-side boxplots of both dates within each quartile

(although see broader comments about alternatives to boxplots).

Response: We considered multiple alternative formats for this figure. Our key comparison is by income status rather than over time (2021 vs 2023), and attempting to overlay both timeframes on the same graph would undermine the primary comparison of interest. However, **Supplementary Table 2** provides data from 2021 and 2023 in a way that is directly comparable.

(5) In general, the color distinction choices do not seem color-blind friendly.

Response: Thank you for your comment. We have used an online tool to select a colour-blind friendly palette and updated figures accordingly.

(6) References needed

There are assertions made in the discussion / conclusions that would benefit from more thorough citations. We list a few examples below.

Lines 217-22: "This variability in vaccine distribution..."

This statement hints at causality, the implication being that variation in income drives variation in the logistics/agreements which in turn explains the observed differences in this analysis. The analysis presented does not include any information on country-level logistics or vaccine-related agreements, so citation of work demonstrating these differences by income and their downstream impacts is needed.

Response: As noted in our response to comment 5 above, we have made amendments to adopt a more cautious approach to causal language. In particular, we now include the following sentence (with multiple relevant citations) discussing the range of contributing factors to global disparities in vaccine access: "Key factors contributing to the broad trends quantified here include: the concentration of vaccine manufacturing capacity in high-income countries; monopolisation of vaccine supplies via bilateral purchasing agreements between high-income countries and vaccine manufacturers^{22,23}; and the recognition by late 2021 that booster doses were necessary to reinforce protection in the context of waning immunity and emerging variants, leading high-income countries to retain available doses for booster campaigns rather than distribute them to LMICs for primary vaccination²⁴."

Lines 242-244: "...robust health data infrastructure; high testing rates for SARS-CoV-2; and VE study platforms and established methodological expertise for other pathogens like influenza."

The current analysis describes differences in timing and numeracy of VE studies by income, however country-level resources, infrastructure, and pre-existing expertise were not directly assessed. While we share the intuition that these factors vary with income, the authors should cite work conclusively showing this.

Response: We have added references to two papers that specifically address barriers facing the conduct of VE studies in resource-constrained settings (Patel et al *Vaccine* 2021; Teerawattananon et al *PLoS One* 2022). We also cite illustrative examples of early VE studies in high-income countries (USA, UK, Denmark, and Israel).

Lines 276-278: "When studies were launched..."

Again without citations this seems to be a statement based on intuition. Has any work been done to investigate the barriers to meeting recruitment targets? It seems that some of the barriers mentioned would only be issues for certain study types and not others (e.g. observation cohorts vs. RCTs).

Response: We have added citation of a mixed-methods evaluation of the African region monitoring vaccine effectiveness (AFRO-MoVE) network (Crawley et al *Vaccine* 2025).

The COVAX program is used to determine the date cutoff for scale-up analyses. It would therefore be prudent to reference the relevant policy information.

Response: The scale-up thresholds are derived from the WHO document titled 'Strategy to Achieve Global Covid-19 Vaccination by mid-2022'. We cite this in the Methods section, and have also added a citation when the scale-up analysis is first introduced in the Results: "We then explored the scale-up of

vaccine implementation based on several coverage milestones specified in the 2021 WHO global COVID-19 vaccination strategy¹⁰."

(7) Line 141: Include citation for the paper from Zambia within the text.

Response: We now cover the first VE study dates earlier in the paragraph, with accompanying citations, as follows: *"VE data for quartiles 3, 2, and 1 were first published on 07 July 2021¹⁶, 27 January 2022¹⁷, and 23 November 2021¹⁸, respectively, although studies from quartile 4 continued to predominate throughout the course of the pandemic (Supplementary Fig. 4)."*

Other clarifications/comments

(8) The quartiles division of GNI is a bit arbitrary. Why not use something like Spearman correlation on date rank vs GNI rank, or Pearson correlation on GNI value? We feel that presenting correlation plots as well as linear and/or rank regression plots would provide a clearer picture of the association between GNI and the individual vaccine metrics.

Response: We have added correlation plots and calculated Spearman's rank correlation coefficients as suggested. See response 1 for further details.

(9) The authors analysis of inequities in vaccine in introduction and scale-up considers the percentage of countries that achieve test coverage/ scale-up by Dec 31, 2021 but then compares the median for time until introduction/ scale-up without a date restriction. A more consistent analysis would be to consider the median time for countries that completed the milestone of scale-up by Dec 31st versus the median time for all countries (including those that did not meet the goal).

Response: Thank you for the suggestion. We agree with the suggestion. We have added relevant lines to **Supplementary Table 1** to include these additional strata, and describe the findings in the text as follows:

- *"Clear discrepancies among GNI quartiles were also apparent when restricted to countries achieving 40% coverage before 31 December 2021 (Supplementary Table 1)."*
- *"Disparities in vaccine scale-up were also apparent when considering GNI as a continuous variable (Spearman's rho -0.73, p <0.001; Fig. 3B), and when restricted to countries achieving 70% coverage by 30 June 2022 (Supplementary Table 1)."*

(10) For VE studies, how are publication dates interpreted? Are the dates for preprints vs published studies comparable? Likely many of the published studies were also preprints at some point - would it make sense to always use preprint date if available?

Response: Thank you - this suggestion aligns with our approach, wherein we used the date of the first preprint version where superseded. We have modified the wording of the Methods to provide further clarification of this: *"We used the earliest pre-print date where applicable for analyses relating to the timing of evidence availability (i.e., taking the publication date of the first pre-print if this was superseded by later versions)."*

(11) All acronyms should be defined upon first use in the main text.

Response: We have modified the text accordingly (e.g. defining GNI at first use).

(12) Clarification is needed in the text itself that GNI is on a per capita basis.

Response: Thanks for this suggestion. We clarify this on first use of the term as follows: *"Among 204 countries with World Bank data on per capita Gross National Income (GNI)..."*

(13) Given the use of the end of COVAX as a cutoff point in the analysis, a brief description of COVAX itself should be included.

Response: COVAX is described in detail in the Discussion section as follows: “COVAX, the vaccine pillar of ACT-A (Access to COVID-19 Tools Accelerator), was launched in April 2020 as a multi-partner collaboration to promote the development, manufacture, fair allocation, and programmatic delivery of COVID-19 vaccines.”

To provide further clarity, we have added a brief definition at first use as follows: “marking the end of COVAX, the multi-partner collaboration to promote the development, manufacture, fair allocation, and delivery of COVID-19 vaccines...”

(14) In general, more discussion of the role that differential age structure may have in driving the observed differences in vaccine coverage across countries is warranted.

Response: As noted in reviewer 1 response 24 above, we now elaborate on the variation in demographic profile across income quartiles in the context of coverage disparities as follows:

“Data on coverage in older adults as of December 2021 were available for 127/203 (63%) countries. Where defined, the threshold used to define older adults varied between 50 and 75 years, with a skew towards younger thresholds in lower-income quartiles (Supplementary Table 3). Among countries reporting, the proportion of the total population accounted for by older adults varied from 25% (17%–27%) in quartile 4 (highest income) to 9% (8–11%) in GNI quartile 1 (lowest income; Supplementary Table 2). Median coverage in older adults was 89% (75–96%), 71% (63–85%), 50% (32–72%), and 10% (4–27%) in GNI quartiles 4 (highest income), 3, 2, and 1 (lowest income), respectively.”

(15) On line 61 the authors state that 194 countries had records of COVID-19 vaccine introduction as of 31 December 2021, but on line 138 it states that 192 reported vaccine introduction. Please clarify why those two numbers differ.

Response: Thank you for highlighting this typo. We have updated the text to ensure the two numbers are consistent (now 203 countries given the changes outlined in response 17 below).

(16) The code runs for the most part, although we did not dig too deeply at this stage point. There is an apparent error in script 4 (selecting a column that doesn't exist) and some of the required packages don't work with the most current version of R. The authors may want to use something like renv to ensure particular versions.

Response: Thank you for reviewing the code. We have fixed the variable naming error in script 4. We appreciate the suggestion to use renv and have implemented this change in the updated repository.

Overall we find the most valuable result of this work for our use is the curated dataset of measures relevant to vaccine program assessment. We think the authors could take two additional steps to make this even more valuable for readers like us:

(17) Efforts to fill in data gaps from other sources. For example, World Bank GNI data was not available for Cuba, but UN data is available. In the several places where there are a small number of missing entries, the authors could improve the resulting dataset by synthesizing more sources.

Response: Thanks for this suggestion. We have updated our analyses to use World Bank GNI data defined by the Atlas metric. This enabled us to extend the analyses to 204 countries (including Cuba) from a single public repository. We have amended the Methods section as follows to clarify this: “Country-level income status was obtained from World Bank estimates of per capita Gross National Income (GNI) based on the Atlas method².”

All text, tables, and figures have been updated accordingly, although there are no substantive changes in the findings.

(18) Provision of a clean data table which for each country shows GNI and all the metrics of vaccine program performance used in this analysis. It seems that Ineq_merged_with_vaccine_data_plus_priority_groups.csv comes closest to this, but it is not well documented. It would be more helpful to have a more user-friendly

version (easier to find, more informative column names / fewer extraneous columns, a data dictionary / schema definition, etc.) highlighted as a key output of this analysis.

Response: Thanks for this excellent suggestion. We agree that a final curated dataset would provide a valuable addition to the csv files generated in our analysis scripts. We have added a curated dataset **Supplementary Data 1**, which contains a relevant set of the final variables alongside a Dictionary tab outlining the sources and interpretation of each column. We cite this as follows: *“Among 204 countries with World Bank data on per capita Gross National Income (GNI) data, 203 had records of COVID-19 vaccine introduction as of 31 December 2021 (Fig. 1A, 2A and Supplementary Table 1; see Supplementary Data 1 for full dataset).”*

An equivalent curated dataset **Supplementary Data 2** is provided for VE data, including one tab for estimates included in the descriptive analysis and one tab for estimates included in the meta-regression.

The code underlying the preparation of these data is available on our Github repository, adding transparency to how any derived variables were calculated from the input databases.

Reviewer #3 (Remarks on code availability):

(19) We attempted to run the code. At this stage, our review was cursory. We did not find that the code "just worked", but it seemed like it would with a bit of troubleshooting.

Response: Thank you for reviewing the code. We have updated the workflow in response to the suggestions above, including the incorporation of renv to enhance reproducibility (as suggested in comment 16). At the point of revised submission, each script of the updated workflow ran successfully.

Reviewing the code itself: it generally seemed organized and clearly written. We did not check that the code is a precisely faithful translation of the described methods in the manuscript at this stage.

Reviewer #4 (Remarks to the Author):

Reviewer #5 (Remarks to the Author):

Response: We thank reviewers #4 and #5 for supporting this constructive set of comments.

Point-by-point response to reviewer comments

Reviewer #1 (Remarks to the Author):

The authors have provided a thorough and (more than) satisfactory response to all the issues raised.

Author responses: We thank the reviewer for their positive response to the revised manuscript.

Reviewer #3 (Remarks to the Author):

Summary

We thank the authors for their detailed consideration of our comments and for addressing the issues that were brought up; these changes have clarified the authors' arguments and strengthened the evidence for those arguments. In particular, the correlation analyses and visualizations using the data without stratification make the trends clear and portable to other analyses. The curated datasets are well-organized, and the README file provided in the GitHub repository makes the analysis workflow clear.

We also find the updated framing of the VE meta-regression analysis clearer and better justified. However, we still have two concerns with this element of the work.

First: overall, there remains language implying the lack of significance provides meaningful insight, e.g. "The only metric in which we did not observe clear global disparities was VE itself." is ambiguously readable as "Global disparity in VE is small." - but that is not the statistical question in the analysis or a logical implication of the non-significance in results. It seems to us that the specific analysis here should be more like confirming that the GNI factor contribution to estimate differences is clearly (i.e. statistically significantly) smaller than some threshold X (which the authors can choose, arguing what seems like a reasonable practical value for worrying about HIC vs LIC differences). We are not meta-regression experts, so cannot recommend specific changes to the analysis here if necessary; if these results somehow already show this conclusion (i.e. the effect of GNI category is identifiably smaller than X threshold at an appropriate significance level), the authors should reframe it as such. If such re-analysis is not possible, then changing the framing to be "the analysis cannot tell if GNI has an effect on VE" seems to be the only recourse.

Author responses: We thank the reviewers for raising these concerns. The meta-regression is set up to explore whether a covariate of interest (in this case GNI quartile) explains a significant proportion of variation in an outcome (in this case VE on a log risk ratio scale) - broadly equivalent to an analysis of variance. In this context, the reviewers are correct that absence of evidence should not be equated with evidence of absence, and we duly acknowledge the lack of statistical power as a key limitation of our study (*"Our meta-regression was underpowered to detect geographic differences in VE due to the limited data available for lower-income quartiles..."*).

Nonetheless, we have taken care to base our conclusions on more than statistical significance. In particular, the pooled VE estimates were broadly similar across GNI quartiles, especially where supported by estimates from multiple countries (as shown in **Fig 5** and **Supplementary Table 6**).

We now highlight the consistency in pooled estimates in the Results as follows: *"Pooled VE estimates were similar across GNI quartiles, with deviations attributable to single-country estimates with wide confidence intervals."*

We also reiterate this point in the Discussion: *"Across three products eligible for inclusion, we observed broadly similar pooled VE estimates across GNI quartiles, with no significant differences in protection against severe COVID-19 according to country income status."*

Overall, we feel that the emerging evidence synthesised in our meta-regression provides clear support for our conclusions regarding the robust and generalisable protection conferred by COVID-19 vaccines across multiple income settings, albeit subject to the limitations highlighted in our Discussion section.

Second, in terms of the technical execution of the meta-regression, we are a bit unclear on precisely what the regression model is. The authors' use of "three-level model" suggests a standard model, but for lay scientist readers not routinely consuming meta-regression, it would be enormously beneficial to see the regression equation(s) to clarify what the levels are. As part trying to figure out what the levels are, we dug into the code - it appears that the model includes a random effect for the country and an observation-level random effect (defined as subgroup = as.factor(1:nrow(.))). Is there supposed to be an observation-level random effect? Why not, say, a study-level random effect? This would be more straightforward to assess with some summary context for readers in discussing the methods, such as would be provided by including the specific equations.

Author responses: We have expanded the Methods to offer more clarity on how our meta-analytic approach was set up to handle the nested structure of the VE data:

"The primary covariate of interest was GNI quartile. We fitted three-level meta-analytic models to obtain pooled VE estimates, incorporating GNI quartile as a moderator variable. Three-level meta-analyses enable nested structures (clusters) in data to be explicitly^{57,66}. We anticipated clustering of individual VE estimates by country, given country-level variation in vaccine programme eligibility and implementation, the timing of waves, and healthcare infrastructure... A moderation test was used to assess whether differences between quartiles were statistically significant."

In this context, country is modelled as a random effect to capture the nested nature of VE estimates by country (with changes in VE according to variant era accounted for by stratification of the analysis). The 'subgroup' variable referred to above simply assigns a unique ID for each individual effect size and is included to specify quantification of the within-cluster (level 2) variance.

We defer to other resources (e.g., Chapter 10 of the openly available meta-analysis tutorial by Harrar et al, 2021) for further details on the structure and specification of three-level meta-analyses, including relevant equations. However, in addition to the explanatory text above, we have clarified the model structure in the **Fig 5** legend as follows: "GNI quartile was included as a moderator in meta-analytic models, while country was included as a random effect to account for the anticipated clustering of estimates at country level."

Equivalent text has been added in the footnotes of **Supplementary Tables 6** and **7**. Together, we feel that these provide sufficient detail to guide readers' interpretation of the findings, and we thank the reviewer for raising this interpretability issue.

We include additional conceptual questions and comments below, but leave it up to the purview of the editor to review these responses and make a final decision.

General clarifications

Line 105: It stands out that the Q2 coverage is so much higher than the rest. Might warrant some discussion.

Author responses: We agree that this result is anomalous and cite it when discussing uncertainty in the coverage estimates for high-risk populations: "*Criteria used to define healthcare workers were not available and age thresholds to define older adults were frequently missing. The high coverage in these populations (e.g., 100% median coverage among healthcare workers in GNI quartile 2 as of December 2023) may reflect an underestimation of the target denominator (defined by International Labor Organization statistics) relative to the coverage numerators captured by WHO reporting systems.*"

Lines 197 – 199: The authors present pooled VE estimates for each GNI quartile. However, only 1 study is used to determine this estimate for GNI quartile 2; is it appropriate to describe as a "pooled" estimate?

Author responses: Although GNI quartile 2 is only represented by a single study, the quartile-level VE estimate and 95% CI are derived from the multi-level meta-regression model that also includes studies from other GNI quartiles. As such, we feel that it is appropriate to refer to it as a 'pooled'

estimate, though we take care to clarify the single underlying study in the text and corresponding supplementary table.

Line 208: Negative VE in the CI for Q2 seems odd. Does the data genuinely support negative efficacy? Alternatively, is it an analytical artefact which is potentially addressable via different assumptions?

Author responses: Negative confidence intervals in VE estimates are not uncommon. They typically reflect uncertainty in the estimated effect size, suggesting that negative effectiveness cannot be ruled out rather than supporting negative effectiveness. Residual confounding can be another potential explanation for this observation (e.g., due to differential health-seeking between vaccinated and unvaccinated individuals). Our pooled estimates reflect the effect sizes and uncertainty in the underlying data, giving rise to confidence intervals that cross 0 in a handful of cases.

Lines 236-238: "The time difference between the initial publication of the VE data in the highest and lowest GNI quartiles was over 9 months." – Do the authors mean to say "median" initial publication?

Author responses: Thank you. This refers to the difference between the date of the first paper published in the highest GNI quartile and the date of the first paper in the lowest GNI quartile. We have now changed "initial" to "first" to clarify: *"The time difference between the first ~~initial~~ publication of the VE data in the highest and lowest GNI quartiles was over 9 months."*

Methods Line 12: Puerto Rico is not a country. Might be helpful to clarify that territories are being treated as their own "country" in the analyses, if this is the case.

Author responses: Thank you for raising this point. We have changed the sentence to avoid the term 'country'. This now reads as follows: *"The Marshall Islands, the Federated States of Micronesia, and Puerto Rico were excluded..."*

We have also added the following clarification to the Methods section, as suggested: *"The analysis includes both countries and territories, but for the purposes of conciseness, we use the term 'country' when referring to individual economies."*

References needed

Lines 230-234: "This variability in vaccine distribution (e.g. cold versus ultra-cold storage)." – This statement could still benefit from having a citation.

Author responses: Thank you for highlighting this. We have added relevant citations covering bilateral agreements (Phelan et al, Lancet 2020), COVAX (Pushkaran et al, BMJ Global Health 2023), and cold-chain considerations (Talbot et al, Vaccine 2025; Lazarus et al, BMJ Global Health 2023).

Figures

Figure 3: It may be helpful to include the correlation coefficient value in the corner of the plot.

Author responses: Done.

Typos/grammar

Affiliation 3 doesn't correspond to any of the authors.

Author responses: Thank you. We have fixed this issue in the revised manuscript.

Line 98: one of the parentheticals includes IQR, the others do not, and earlier similar quantitative results also do not include the IQR annotation; whatever the convention for stating or not stating the confidence interval definition, it should be applied consistently.

Author responses: Thank you for highlighting this inconsistency. We have removed 'IQR' from the parentheses in question and now signpost the use of IQRs with each reference to the median metric. For example: "... the median (IQR) time from vaccine introduction to this coverage threshold was 190 (160–209), 239 (182–274)..."

In the paragraph from line 203 in which 95% CIs rather than IQRs are presented in parentheses, we now clarify this at each usage to avoid any potential ambiguity. For example: "Pooled VE estimates for severe COVID-19 due to the Omicron variant were 74% (95% CI 57–84%, 18 studies, 6 countries), 67% (95% CI 38–82%, 5 studies, 4 countries), and 49% (95% CI 49–87%, 1 study, 1 country) for GNI quartiles 4, 3, and 2, respectively..."

We are happy to defer to the editorial team for further guidance on preferred practice here.

Line 100: Make "Fig. 3B" bold.

Author responses: All bold text has now been removed, as per editorial guidance.

Line 135: There is a stray comma after em dash.

Author responses: The comma has been removed.

Line 160: Make "Fig. 3C" bold.

Author responses: All bold text has now been removed, as per editorial guidance.

Lines 182-184: "We selected across multiple GNI quartiles." – Is this sentence grammatically correct?

Author responses: We have now added "with" to improve clarity: "We selected product- and variant-specific VE estimates relating to adults (of any age group), with complete homologous primary series vaccination, and with representation across multiple GNI quartiles."

Line 201-2: "...series vaccination by was 25..." – This sentence also seems grammatically incorrect; there may be a word missing.

Author responses: Thank you. We have fixed this in the revised manuscript. "The median (IQR) for the upper limit of follow-up (the maximum potential duration of follow-up among study participants) after primary series vaccination ~~by~~ was 25 (21–35) and 28 (16–48) weeks for GNI quartiles 4 and 3, respectively, and 38 weeks for the single estimate from GNI quartile 2."

Methods Line 91: "package" should be plural.

Author responses: Thank you. We have fixed this in the revised manuscript.

Reviewer #3 (Remarks on code availability):

Yes, after some markdown troubleshooting.

Author responses: Thank you for reviewing the code and for the positive appraisal of the Github repository above.